

# Extracting quantum-critical properties from directly evaluated enhanced perturbative continuous unitary transformations

**L. Schamriß**[⋆]**, M. R. Walther**[†] **and K. P. Schmidt**[‡]

Department of Physics, Friedrich-Alexander-Universität Erlangen-Nürnberg (FAU),
Staudtstraße 7, Germany

⋆ lukas.schamriss@fau.de , † matthias.walther@fau.de , ‡ kai.phillip.schmidt@fau.de

## Abstract

Directly evaluated enhanced perturbative continuous unitary transformations (deep-CUTs) are used to calculate non-perturbatively extrapolated numerical data for the ground-state energy and the energy gap. The data coincide with numerical evaluations of the truncated perturbative series and provide robust extrapolations beyond the perturbative regime. We develop a general scheme to extract quantum-critical properties from the deepCUT data based on critical scaling and a strict correspondence between the truncation used for deepCUT and the length scale of correlations at the critical point. We apply our approach to transverse-field Ising models (TFIMs) as paradigmatic systems for quantum phase transitions of various universality classes depending on the lattice geometry and the choice of antiferromagnetic or ferromagnetic coupling. In particular, we focus on the quantum phase diagram of the bilayer antiferromagnetic TFIM on the triangular lattice with an Ising-type interlayer coupling. Without a field, the model is known to host a classically disordered ground state, and in the limit of decoupled layers it exhibits the 3d-XY 'order by disorder' transition of the corresponding single-layer model. Our starting point for the unknown parts of the phase diagram is a high-order perturbative calculation about the limit of isolated dimers where the model is in a gapped phase.

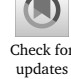

# 1   Introduction

Investigating quantum criticality with perturbative methods usually involves a non-perturbative extrapolation in order to account for the truncated long-range correlations in the critical regime. Indeed, there is a variety of series expansion methods [1–6] which allow to efficiently calculate high-order series expansions in the thermodynamic limit, typically using graphs and linked-cluster expansions. The standard approach for the extrapolation are dlog-Padé approximants [7] whose Taylor series is exactly the perturbative series up to the order in which the perturbative result is available. The higher orders in their expansion are designed to capture critical behavior like power laws.

Moreover, there exists a variety of methods which are considered non-perturbative but share the property of dlog-Padé approximants that they recover the perturbative series up to a specific order and add extrapolated higher orders. First of all, there is the numerical linked-cluster expansion (NLCE) [8–10] which exploits exact diagonalization in combination with a linked-cluster expansion for the extrapolation. In addition, also the density matrix renormalization group as well as the hybrid variational quantum eigensolver have been used as algorithms within the NLCE scheme to perform calculations on the linked clusters [11,12]. The recently introduced projective cluster-additive transformation (PCAT) allows to extend the range of applicability from ground states to excitations [6]. Both is as well achieved by the graph-theory based continuous unitary transformation (gCUT) [13] which however also is known to have problems with convergence [14], as well as NLCE using PCAT. In the context of gCUT, a general approach to extract quantum criticality from non-perturbative numerical data based on dlog-Padé extrapolations has been introduced [15].

Further methods based on flow equations are also the continuous similarity transformation (CST) for gapless and gapped phases [16, 17] and the self-similar continuous unitary transformation (sCUT) [18]. CST operates in the momentum space and applies a truncation in the scaling dimension which is tuned to capturing critical properties. It has been applied successfully to a transition in the easy-axis antiferromagnetic XXZ model [19], but no general way of analysis for quantum criticality has been established. The analysis of results from sCUTs have been found to be intricate since the truncation in real space must be designed manually which limits the control over the truncation errors. DeepCUT shares the property to be a non-perturbative method capturing a perturbative series. If explicitly only the series is calculated in the framework, the transformation is called enhanced perturbative continuous unitary transformation (epCUT) and is a direct extension of perturbative continuous unitary transformation (pCUT) [3, 4] since it cures the limitation of pCUT to equidistant spectra in the unperturbed Hamiltonian. However, in contrast to pCUT, the flow equation in that generalized case must be solved numerically and the linked-cluster property of the transformation has so far not been used for a linked-cluster expansion. This makes the method significantly more expensive computationally such that the accessible perturbative orders are lower than for pCUT, NLCE or PCAT. Recently, pcst$^{++}$ has been introduced [20] as a new method which also avoids the restriction of pCUT to equidistant spectra. In contrast to epCUT, the flow equation is still solved algebraically as for pCUT and as an additional feature, pcst$^{++}$ can also be used to treat open systems. The strength of epCUT lies in its non-perturbative extrapolation achieved by deepCUT. The algorithmic requirements and the computational cost are almost the same as for epCUT. However, it has been shown to be a quantitatively robust extrapolation as compared to dlog-Padé extrapolations [21] and solves the issue of the truncation not being well-defined for sCUTs.

DeepCUT has widely been applied [22–24] to investigate properties of models in parameter regimes where the perturbative series is not meaningful anymore. In most cases one-dimensional models are considered, such as the one-dimensional TFIM [25], and to our knowledge no efforts have been made to access universal properties of quantum phase transitions. We only are aware of one application of deepCUT to a two-dimensional model [26] which neither has addressed quantum criticality. In this article, we will fill this gap and investigate for various two-dimensional TFIMs, how the universality classes which the models fall into can be discriminated with deepCUT. TFIMs are well-suited, paradigmatic systems displaying quantum phase transitions of various universality classes depending on the lattice geometry and the choice of antiferromagnetic or ferromagnetic coupling. A focus is laid on the geometrically frustrated bilayer TFIM with antiferromagnetic Ising interactions on the triangular lattice with an Ising-type interlayer coupling. In the absence of a field, this model is known to host a classically disordered ground state, and in the limit of decoupled layers it exhibits the 3d-XY *'order by disorder'* transition of the corresponding frustrated single-layer model. We consider the limit of isolated dimers as the starting point for our deepCUT calculations allowing to map out the quantum phase diagram of the systems.

The article is organized as follows. In Sec. 2 we introduce the physical properties and notations of the TFIMs investigated in this work including the scaling properties of second-order quantum phase transitions. Sec. 3 contains all relevant information about the deepCUT method. The iterative scheme to detect quantum criticality based on the energy gap with the deepCUT method is presented in Sec. 4 while a sanity check for the correspondence of truncation and length scale is given in Sec. 5. The iterative scheme is then applied to the TFIM bilayer on the triangular lattice and corresponding results are contain in Sec. 6. We summarize and conclude our work in Sec. 7.

Table 1: Values for the quantum critical points $J_c/h$ and critical exponents of the TFIM on the triangular lattice with antiferromagnetic and ferromagnetic coupling.

| triangular TFIM | antiferromagnetic | ferromagnetic |
|---|---|---|
| universality class | 3d XY | 3d Ising* |
| critical point $J_c/h$ | 0.610(4) [27] | 0.20973(2) [7] |
| exponent $z$ | 1 | 1 |
| exponent $\nu$ | 0.67175(10) [28] | 0.629971(4) [29] |
| exponent $\alpha$ | −0.01526(30) [28] | 0.110087(12) [29] |

## 2 Quantum criticality in Ising models

### 2.1 Transverse-field Ising models

We consider a nearest-neighbor spin-1/2 TFIM described by the Hamiltonian

$$H_I = J \sum_{\langle i,j \rangle} \sigma_i^x \sigma_j^x + h \sum_i \sigma_i^z, \tag{1}$$

where $J$ is the Ising coupling constant, $h$ is the transverse-field strength, and $\sigma_i^{x,y,z}$ are standard Pauli matrices acting on site $i$. The Hamiltonian can be rewritten in second quantization in terms of hardcore bosons with creation (annihilation) operators $b_i^\dagger$ ($b_i$) and the commutation relation

$$\left[ b_i, b_j^\dagger \right] = \delta_{ij} \left( 1 - 2n_i \right). \tag{2}$$

The operator $n_i \equiv b_i^\dagger b_i$ with eigenvalues $0, 1$ counts the number of quasi-particles at site $i$. Up to a constant shift in the energy, the Hamiltonian in terms of hardcore bosons reads

$$H_I = 2h \sum_i n_i + J \sum_{\langle i,j \rangle} \left( b_i^\dagger + b_i \right)\left( b_j^\dagger + b_j \right). \tag{3}$$

TFIMs can host a variety of phases and quantum phase transitions which make them an interesting object of investigation. One workhorse system in this work is the TFIM on the triangular lattice with ferromagnetic ($J < 0$) and antiferromagnetic ($J > 0$) coupling. For both cases the respective quantum phase diagram has been explored numerically [7, 27, 30, 31] and therefore can be used to benchmark the performance of our deepCUT approach in the vicinity of quantum-critical points. The ferromagnetic TFIM on the triangular lattice displays a quantum phase transition between the high-field polarized phase and the low-field $\mathbb{Z}_2$ symmetry-broken phase in the 3d Ising universality class. In contrast, the highly frustrated antiferromagnetic TFIM on the triangular lattice displays a 3dXY transition due to an emergent $O(2)$-symmetry at the quantum critical point [32]. Here the ground state at infinitesimal magnetic fields is given by a clock-ordered state induced by a *'order by disorder'* scenario. Literature values for the quantum critical points and critical exponents are listed in Tab. 1. The numerical values for the exponents are taken from conformal-bootstrap studies [28, 29].

### 2.2 Frustrated transverse-field Ising bilayer

Next we introduce the frustrated TFIM bilayer on the triangular lattice for which the phase diagram has not been explored to the best of our knowledge. To this end we consider the stacking of two TFIMs on the triangular lattice so that both layers are coupled by an antiferromagnetic nearest-neighbor Ising interaction $J_\perp$. A spin on site $\mu$ and layer $i$ is therefore represented

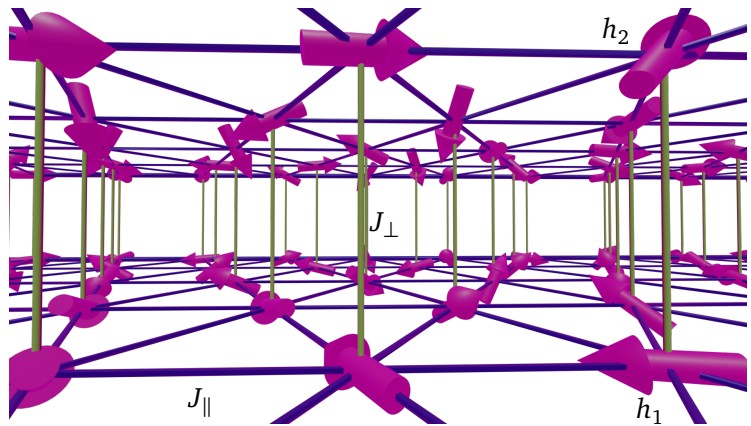

Figure 1: Illustration of the frustrated transverse-field Ising bilayer. Two triangular TFIMs with intralayer coupling $J_\parallel$ along blue lines and fields $h_{1,2}$ on layers $1, 2$ are stacked and coupled via the interlayer coupling $J_\perp$ along green lines.

by $\sigma_{\mu i}^{x,y,z}$. The couplings of the frustrated TFIM bilayer are illustrated in Fig. 1 and the model is described by the Hamiltonian

$$H = J_\parallel \sum_{i=1,2} \sum_{\langle \mu,\nu \rangle} \sigma_{\mu i}^x \sigma_{\nu i}^x - J_\perp \sum_\mu \sigma_{\mu 1}^x \sigma_{\mu 2}^x + \sum_{\mu,i} h_i \sigma_{\mu i}^z . \tag{4}$$

$J_\parallel > 0$ is the interaction strength of the intra-layer coupling inducing geometric frustration while $J_\perp > 0$ is the inter-layer coupling. The sign of $J_\perp$ can be fixed without loss of generality since it can be absorbed into the relative orientation of the spins in layer 1 and 2. The field strengths coupling to the spins in layer 1 and 2 is denoted by $h_{1,2}$. For the case $h_1 = h_2$ the model is symmetric under the permutation of the layers.

Three limiting cases in the parameter space of the model give insights into the quantum phase diagram of the model. First, in the limiting case of vanishing inter-layer coupling, $J_\perp = 0$, the model hosts two copies of the frustrated TFIM on the triangular lattice and hence, the phase diagram contains a quantum critical point in the 3d-XY universality class. The phases on either side of the critical point are gapped and hence are expected to extend into the region of non-vanishing inter-layer coupling in the phase diagram. Second, in the case of vanishing fields, $h_{1,2} = 0$, the model inherits the extensive ground-state degeneracy from the single-layer TFIM since intra- and inter-layer couplings commute. An infinitesimal field induces an *'order by disorder'* scenario and the same quantum phase as for the single-layer model. Third, in the case of decoupled dimers ($J_\parallel = 0$), the ground state is unique as long as the field terms do not vanish and the state is adiabatically connected to the polarized state in the high-field limit of the decoupled layers. Hence, in the limit $J_\perp \gg h_{1,2}, J_\parallel$ with intra-layer coupling only, which is the boundary between cases two and three, there must be a transition between the polarized and the clock-ordered phase for various ratios of infinitesimal $J_\parallel$ and $h_{1,2}$. This transition can be understood in the context of a perturbative model in the low-energy physics of the dimers $\sigma_{\mu 1}^x \sigma_{\mu 2}^x$ using a mapping of the spin degrees of freedom onto a hardcore boson $b_\mu, b_\mu^\dagger$ per site, accounting for the high-energy scale, and a pseudo spin $\tau_\mu^{x,y,z}$ for the low-energy scale [33]. Formally, this mapping is given by

$$\begin{aligned}
\sigma_{\mu 1}^z &= \tau_\mu^x (b_\mu^\dagger + b_\mu), & \sigma_{\mu 1}^x &= \tau_\mu^z, \\
\sigma_{\mu 2}^z &= b_\mu^\dagger + b_\mu, & \sigma_{\mu 2}^x &= \tau_\mu^z (1 - 2 b_\mu^\dagger b_\mu).
\end{aligned} \tag{5}$$

One can check that indeed the particle number operator $n_\mu \equiv b_\mu^\dagger b_\mu$ encodes the energy of the dimer $\sigma_{\mu 1}^x \sigma_{\mu 2}^x = 1 - 2n_\mu$. We rewrite the Hamiltonian exactly as

$$H = 2J_\perp \sum_\mu n_\mu + J_\parallel \sum_{\langle \mu \nu \rangle} \tau_\mu^z \tau_\nu^z \big(1 + (1 - 2n_\mu)(1 - 2n_\nu)\big) + \sum_\mu (h_1 \tau_\mu^x + h_2)(b_\mu^\dagger + b_\mu), \quad (6)$$

in terms of hardcore bosons and pseudo spins. In the low-field and low-$J_\parallel$ limit the energy scale of this interaction is dominant and the low-energy physics driving the phase transition lies within the zero-hardcore-boson sector of the Hilbert space which is described by the pseudo spins. The effective Hamiltonian $H_0^{(3)}$ in third-order perturbation theory in $h_{1,2}$ and $J_\parallel$ under the condition $n_\mu = 0$ reads

$$H_0^{(3)} = -N \frac{h_1^2 + h_2^2}{2J_\perp} + \underbrace{\left(2J_\parallel - \frac{3}{2} \frac{(h_1^2 + h_2^2)J_\parallel}{J_\perp^2}\right)}_{\tilde{J}} \sum_{\langle \mu \nu \rangle} \tau_\mu^z \tau_\nu^z - \underbrace{\frac{h_1 h_2}{J_\perp}}_{\tilde{h}} \sum_\mu \tau_\mu^x. \quad (7)$$

It describes the physics within the low-energy degrees of freedoms introduced by an infinitesimal field and corresponds to an effective single-layer TFIM with coupling $\tilde{J}$ and field $\tilde{h}$ on the triangular lattice. This implies that in the corner $J_\perp \gg h_{1,2}, J_\parallel$ of the phase diagram a critical line starts, which separates a polarized and an ordered phase by a 3d-XY transition. Given the literature value $x_c := J_c/h = 0.610(4)$ for the single-layer model, we can express the onset of the critical line in the phase diagram as

$$J_{\parallel, c} = x_c \frac{h_1 h_2}{2J_\perp} + \mathcal{O}(h^4), \quad (8)$$

where the cubic order exactly vanishes. It will be used to estimate the convergence of the critical line obtained from deepCUT data in Sec. 6.

Next we transform the model into an energetically equivalent single-layer model with three instead of one excitation per dimer. This is a requirement for the application of deepCUT and is achieved by defining quasi-particles as eigenstates of the dimers. The Hamiltonian of a single dimer reads

$$H_{\text{dimer}} = J_\perp \sigma_1^x \sigma_2^x + \sum_{i=1}^2 h_i \sigma_i^z. \quad (9)$$

For diagonalizing it, we leverage the spin-flip symmetry induced by the conserved quantity $\sigma_1^z \sigma_2^z$. A basis respecting that symmetry is given by

$$\begin{pmatrix} |\rightarrow\rightarrow\rangle + |\leftarrow\leftarrow\rangle \\ |\rightarrow\rightarrow\rangle - |\leftarrow\leftarrow\rangle \\ |\rightarrow\leftarrow\rangle - |\leftarrow\rightarrow\rangle \\ |\rightarrow\leftarrow\rangle + |\leftarrow\rightarrow\rangle \end{pmatrix}, \quad (10)$$

in terms of the eigenstates of the Pauli $x$-operators, $\sigma^x |\rightarrow\rangle = |\rightarrow\rangle$ and $\sigma^x |\leftarrow\rangle = -|\leftarrow\rangle$. Introducing the symmetric and antisymmetric combination of the fields $h_\pm := h_2 \pm h_1$, the Hamiltonian in that basis reads

$$H_{\text{dimer}} = \begin{pmatrix} -J_\perp & 0 & 0 & h_+ \\ 0 & -J_\perp & h_- & 0 \\ 0 & h_- & J_\perp & 0 \\ h_+ & 0 & 0 & J_\perp \end{pmatrix}. \quad (11)$$

It decouples into two $2 \times 2$ subblocks which can be easily diagonalized using the following notation

$$\epsilon_\pm = \sqrt{J_\perp^2 + h_\pm^2}, \qquad u_\pm = \sqrt{\frac{1}{2}\left(1 - \frac{J_\perp}{\epsilon_\pm}\right)}, \qquad v_\pm = \sqrt{\frac{1}{2}\left(1 + \frac{J_\perp}{\epsilon_\pm}\right)},$$

$$\Gamma_\pm = v_+ v_- \pm u_+ u_-, \qquad \Xi_\pm = -(v_+ u_- \pm u_+ v_-). \tag{12}$$

The resulting local eigenenergies are $(-\epsilon_+, -\epsilon_-, \epsilon_-, \epsilon_+)$. The perturbation in the respective eigenbasis is found by transforming the operators

$$\sigma_1^x = \begin{pmatrix} 0 & \Gamma_+ & \Xi_- & 0 \\ \Gamma_+ & 0 & 0 & -\Xi_- \\ \Xi_- & 0 & 0 & \Gamma_+ \\ 0 & -\Xi_- & \Gamma_+ & 0 \end{pmatrix}, \qquad \sigma_2^x = \begin{pmatrix} 0 & \Gamma_- & \Xi_+ & 0 \\ \Gamma_- & 0 & 0 & \Xi_+ \\ \Xi_+ & 0 & 0 & -\Gamma_- \\ 0 & \Xi_+ & -\Gamma_- & 0 \end{pmatrix}. \tag{13}$$

Whenever the fields $h_{1,2}$ do not vanish, a unique ground state in each dimer can be identified as the vacuum state in the quasi-particle description. The three remaining states per dimer are local excitations and are expressed by two independent quasi-particle flavors with hardcore-bosonic commutation relations. For this sake, we set the vacuum energy to zero and introduce the excitation energies

$$\epsilon_1 := \epsilon_+ - \epsilon_-, \qquad \epsilon_2 := \epsilon_+ + \epsilon_-, \qquad \epsilon_3 := 2\epsilon_+. \tag{14}$$

In the dimer eigenbasis, the creation operators act on the vacuum $|0\rangle$ as

$$b_1^\dagger |0\rangle = \begin{pmatrix} 0 \\ 1 \\ 0 \\ 0 \end{pmatrix}, \qquad b_2^\dagger |0\rangle = \begin{pmatrix} 0 \\ 0 \\ 1 \\ 0 \end{pmatrix}, \qquad b_1^\dagger b_2^\dagger |0\rangle = \begin{pmatrix} 0 \\ 0 \\ 0 \\ 1 \end{pmatrix}. \tag{15}$$

The Hamiltonian (4) can then be rewritten exactly in second quantization as

$$H = \sum_\mu \sum_{a=1,2} \epsilon_a n_{a\mu} + J_\parallel \sum_{\langle \mu, \nu \rangle} \left\{ \left[ \Gamma_+ b_{1\nu}^\dagger + \Xi_- (1 - 2n_{1\nu}) b_{2\nu}^\dagger + \text{h.c.} \right] \cdot \left[ \nu \to \mu \right] \right.$$

$$\left. + \left[ \Gamma_- (1 - 2n_{2\nu}) b_{1\nu}^\dagger + \Xi_+ b_{2\nu}^\dagger + \text{h.c.} \right] \cdot \left[ \nu \to \mu \right] \right\}, \tag{16}$$

where $\nu \to \mu$ is a substitute for the previous term on site $\mu$ instead of $\nu$. A more detailed form of the Hamiltonian and a simplified form for $h_1 = h_2$ is given in App. A. Note that all numerical results presented in this work are obtained for the latter symmetric case.

## 2.3 Scaling properties of second-order quantum phase transitions

In the vicinity of a second-order quantum phase transition driven by the model parameter $g$ at the critical point $g_c$, the length scale $\xi$ of the quantum fluctuations diverges and the properties of the system are usually described by universal power laws which are determined by the universality class of the transition. The behavior of the correlation length $\xi$ is described by the power law $\xi \sim |g - g_c|^{-\nu}$ with the universal critical exponent $\nu$. The behavior of the energy gap $\Delta$ at the quantum critical point $g_c$ can be derived from the scaling form

$$\Delta(|g - g_c|) = \xi^{-z} \Delta\left(\xi^{\frac{1}{\nu}} |g - g_c|\right), \tag{17}$$

which has been considered by Hamer [34] using the dynamical critical exponent $z$. One finds $\Delta \sim \xi^{-z}$ for the scaling at the critical point. For discriminating universality classes, $\nu$ can be determined based on the first derivative of the gap

$$\partial_g \Delta \sim |g - g_c|^{z\nu-1} \sim \xi^{-z+\frac{1}{\nu}}\,, \tag{18}$$

which has also been pointed out by Hamer [34]. The closing of the gap $g_{(\Delta=0)}$ gives access to the critical point. Its asymptotic scaling with respect to the length scale is given by

$$|g_{(\Delta=0)} - g_c| \sim \xi^{-\frac{1}{\nu}}\,, \tag{19}$$

according to the scaling form in Eq. 17. Additionally, the heat capacity which is linked to the second derivative of the ground-state energy, is given in App. D since it gives access to the critical exponent $\alpha$. In order to link the introduced scaling properties to deepCUT, the correspondence

$$\xi \propto n\,, \tag{20}$$

between truncation to the perturbative order $n$ and length scale is assumed, motivated by the property of perturbation theory in lattice systems with short-range interactions, that the spatial extent of the relevant quantum fluctuations increases linearly with the perturbative order $n$. In Sec. 5, as also observed for the one-dimensional TFIM [25], we give numerical evidence for this correspondence.

# 3 Directly evaluated enhanced perturbative continuous unitary transformations

## 3.1 General derivation

The directly evaluated enhanced perturbative continuous unitary transformation (deepCUT) relies on the numerical solution of a non-perturbative flow equation in the space of Hamiltonian operators in a perturbatively truncated finite basis. A detailed derivation is given in Ref. [21] which we review in this section. We start the derivation of deepCUT by splitting the Hamiltonian $H = H_0 + xV$ into an exactly solvable part $H_0$ and a perturbation $V$. In the context of flow equations, a one-parameter family of Hamiltonians $H(l)$ with $0 \le l < \infty$ is considered. $H(0) := H$ is the starting condition for the flow equation

$$\partial_l H(l) = [\hat{\eta}[H(l)], H(l)]\,, \tag{21}$$

defining $H(l)$ for all finite values of $l$ and the desired block-diagonal effective Hamiltonian $H_{\text{eff}} := H(\infty)$. The only choice to be made is the form of the superoperator $\hat{\eta}$ which by standard is defined in terms of the basis of normal-ordered monomials (monoms) $A_i = b_{i_1}^{(\dagger)} \dots b_{i_n}^{(\dagger)}$ of second-quantization operators as

$$\hat{\eta}[A_i] = \text{sign}(\Delta E_i) A_i\,, \tag{22}$$

where $\Delta E_i$ is the change of energy with respect to $H_0$ when $A_i$ is applied to a state [4]. This generator is called quasi-particle (qp) conserving generator and decouples all qp sectors. If however, as for many applications, only the ground-state energy and the energy gap are desired, it is sufficient to decouple only the 0-qp block for the ground state or the 0-qp and 1-qp block for the energy gap from all others. This idea has been put forward by the introduction of the 0:$n$ and 1:$n$ generators [35]. The advantages are a performance boost from the decreased

basis size and a potentially extended range of convergence of the deepCUT. Importantly, we have sign(0) = 0 such that diagonal monoms are not suppressed by the flow. To set up the flow equation (21), the Hamiltonian is written in the basis of monomials as $H = \sum_i h_i A_i$ with coefficients $h_i$. The flow equation expanded in this basis reads

$$\sum_i \partial_l h_i(l) A_i = \sum_{ijk} h_j(l) h_k(l) [\hat{\eta}[A_j], A_k].$$ (23)

The commutator can as well be expanded yielding

$$[\hat{\eta}[A_j], A_k] = \sum_i D_{ijk} A_i,$$ (24)

by introducing the symbols $D_{ijk}$. This allows us to set up the flow equation in the form

$$\partial_l h_i(l) = \sum_{jk} D_{ijk} h_j(l) h_k(l),$$ (25)

which can be solved numerically if the basis is finite. The effective Hamiltonian with now decoupled quasi-particle blocks is directly read off as

$$H_{\text{eff}} = \sum_i h_i(\infty) A_i.$$ (26)

The results are the coefficients $h_i(\infty)$ of the monomials. For example, the monom $\mathbb{1}$ has the vacuum energy per site as its coefficient since it is the only normal-ordered monom which does not vanish when it acts on the vacuum state. The energy gap is obtained from the Fourier transform of the monomials $\sum_i C_i a_i^\dagger a_{i+\delta}$ describing the hopping of a single excitation, analogously.

### 3.1.1 Perturbative expansion and truncation of the flow equation

In order to calculate the coefficients of a perturbative series numerically stable, the flow equation (21) must be expanded. We start with the perturbative expansion of the Hamiltonian $H^{(m)}(l) = \sum_i f_i^{(m)}(l) A_i$ for any order $m$. In terms of the newly introduced coefficients $f_i^{(m)}$, the flow equation reads

$$\partial_l f_i^{(m)}(l) = \sum_{jk} \sum_{p+q=m} D_{ijk} f_j^{(p)}(l) f_k^{(q)}(l).$$ (27)

It can be solved numerically in order to gain the series expansion of any matrix element of the effective Hamiltonian, such as the vacuum energy per site

$$\langle 0|H_{\text{eff}}|0\rangle = \sum_{m=0}^n x^m f_i^{(m)}(\infty) + \mathcal{O}(x^{n+1}),$$ (28)

for the monom $i$ given by $A_i = \mathbb{1}$. The choice of a targeted quantity specifies for which matrix elements of the effective Hamiltonian the series expansion is required. For any such targeted quantity and any order $n$, the contributions $D_{ijk}$, which are strictly necessary to obtain the correct expansion, can be identified and all others can be neglected. In particular, this also reduces the basis $\{A_i\}$ to the minimal necessary basis if those monomials are dropped whose coefficients do not influence the targeted ones via the reduced set of contributions $D_{ijk}$. For the evaluation of a series expansion in the case of epCUT, the reduction of the equation is merely an optional optimization reducing the size of the differential equation system which must be solved numerically. We review the technical details how to efficiently generate the minimal necessary basis in App. B.

### 3.1.2  Direct evaluation of the flow equation

In contrast, for deepCUT, the reduced set of contributions defines the truncation of the flow equation and therefore the outcome. The flow equation (21) is set up as in Eq. 25 but with strictly necessary contributions only. Matrix elements can be obtained directly as for example the vacuum energy with $A_i = \mathbb{1}$

$$\langle 0|H_{\text{eff}}|0\rangle = h_i(\infty) + \mathcal{O}(x^{n+1}). \tag{29}$$

The error in the latter equation is expected to be significantly smaller than for epCUT. The coefficients $h_i(\infty)$ in general cannot be written as a series up to order $n$, but rather yield an infinite series if tailored. Hence, one finds

$$\sum_{m=0}^{n} x^m f_i^{(m)}(\infty) + \mathcal{O}(x^{n+1}) = h_i(\infty), \tag{30}$$

showing that the deepCUT result contains all information about the series from epCUT and additionally, a non-perturbative extrapolation resembling the true value for $h_i$ in the effective Hamiltonian. This extrapolation depends on the truncation of the differential equation system. Krull et al. have shown that the truncation for deepCUT extends the range of convergence as compared to the series from epCUT [21].

### 3.2  Numerical solution of flow equations

The numerical solution of the differential equations is found using the dopri5 solver [36] from the scipy library for python. The break-off condition detecting the convergence in the limit of infinite flow parameter is a drop of the residual off-diagonality (ROD) defined as

$$\sqrt{\sum_{\hat{\eta}(A_i)\neq 0} h_i(l)^2} < \epsilon, \tag{31}$$

below the threshold $\epsilon$, where the sum contains all monomials which are aimed to be eliminated according to the selected generator. We terminate the flow at $\epsilon = 1 \times 10^{-9}$ in the respective units of energy. We note that this is close to the limit given by double precision in the presence of $1 \times 10^{6}$ monoms resulting potentially in a cumulative numerical error. In addition, we observed that the relevant physical quantities are fully converged with sufficient accuracy. The relative tolerance in the solver is set to $1 \times 10^{-6}$. For the detection of a divergence in the flow, which can be present in deepCUT, the minimal ROD which occurs in the flow from 0 to $l$ is tracked. If the ROD exceeds the minimum at some point during the flow by a factor of $1 \times 10^{8}$ or more, the deepCUT is considered to be non-convergent for the respective parameters.

## 4  Iterative scheme to detect quantum criticality based on the energy gap

The correspondence of perturbative order and length scale in Eq. 20 serves as the starting point to analyze quantum criticality based on the energy gap. The critical exponent $\nu$ describes the asymptotic closing of the gap according to Eq. 19 and the asymptotic behavior of the derivative of the energy gap according to Eq. 18. We use the latter to extract $\nu$ via a linear least-squares fit in a log-log plot of the deepCUT data versus the perturbative order. For the single-layer models energies from deepCUT are always given in units of $\sqrt{J^2 + h^2}$. The divergence of the derivative is expected to follow a power law with the exponent $-z + 1/\nu$, which corresponds to

Table 2: Iterations of the newly developed scheme to detect quantum-critical properties based on deepCUT data. The model is the TFIM on the triangular lattice with antiferromagnetic and ferromagnetic coupling.

| | antiferromagnetic | | ferromagnetic | |
|---|---|---|---|---|
| Iteration | $\nu$ | $J_c$ | $\nu$ | $J_c$ |
| 1 | 1 | 0.592 | 1 | 0.2035 |
| 2 | 0.71 | 0.607 | 0.665 | 0.2088 |
| 3 | 0.69 | 0.608 | 0.628 | 0.2094 |
| 4 | 0.69(3) | 0.608(3) | 0.621 | 0.2095 |
| 5 | | | 0.621(3) | 0.2095(2) |
| literature | 0.67 | 0.610(4) | 0.630 | 0.2097 |

the slope of the fit and is used to calculate $\nu$ by assuming $z = 1$. The results are shown in Fig. 2a for the antiferromagnetic TFIM on the triangular lattice and in Fig. 2c for the corresponding ferromagnetic case. We add the standard error of the linear fits in brackets. The reached accuracy is sufficient for discriminating the corresponding universality classes. This underlines the usefulness of the scaling-based approach to quantum criticality in the context of deepCUT, if the critical point is known. Using the asymptotic bahavior of the closing of the gap, we can however also extract the critical point if the value of $\nu$ is given. By plotting the closing of the gap against $n^{-1/\nu}$ as done in Figs. 2b and 2d using the literature values for $\nu$ according to the respective universality classes, we expect an asymptotically linear decay in the limit $n^{-1/\nu} \to 0$ according to Eq. 19. Hence, a linear fit can be used as an unbiased extrapolation towards infinite perturbative order. The resulting extrapolated value $J_c/h = 0.609(3)$ for the frustrated case shown in Fig. 2b is well within the errors of the quantum Monte Carlo literature value of 0.610(4) [27]. For the ferromagnetic model in Fig. 2d it agrees well with the results from series expansions obtained by Hamer [7]. In contrast to the frustrated TFIM, both scaling behaviors closely follow their power laws if the lowest orders are neglected, giving numerical evidence for the expected scaling relations. The critical point of the frustrated model could be exrapolated best by considering means of even and odd orders which are shown as crosses in Fig. 2b.

Fig. 2 shows that either the critical point or the critical exponent $\nu$ can be found precisely if the corresponding other quantity is known. This motivates an iterative technique: Starting with the estimate $\nu = 1$ one can alternatingly feed the the latest value for $J_c$ ($\nu$) to the fit for $\nu$ ($J_c$). The results for the TFIMs on the triangular lattice are given in Tab. 2 for both cases. The method converges quickly and gives results for the critical point and the critical exponent $\nu$ at similar precision as dlog-Padè extrapolation. No literature values apart from $z = 1$ need to be known beforehand. Hence, the universality can be determined with high certainty.

Here, for performing the scheme a set of data points $\Delta^{(n)}(J_i)$ is generated for all available orders $n$ at discretized couplings $J_i$ which are chosen equidistantly around the presumed critical point and which are required to cover the closing of the gap in all orders, importantly. From that the closing of the gap $J_0^{(n)}$ can be extracted via linear interpolation, and the first guess for $J_c$ is extrapolated assuming $\nu = 1$ in the first iteration. We find, that the result of the scheme does not depend on that choice. Then the coupling $J_i$ closest to $J_c$ is chosen for the extrapolation fit of the first non-trivial value of $\nu$ in the second iteration. Eventually in later iterations, the change in the guess for the critical point may be smaller than the spacing of the coupling values $\delta$ which will result in the same $\nu$ as in the previous step. Hence, the discretization of $J_i$ induces a discretization of the accessible values for $\nu$. The critical point

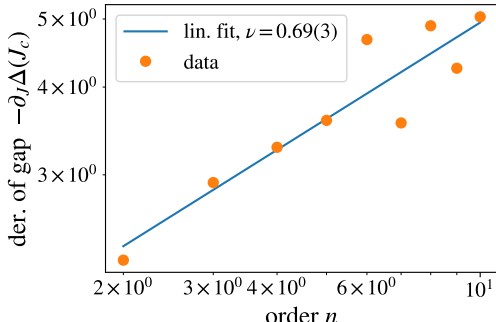

(a) Antiferromagnetic triangular TFIM, extrapolation of the exponent.

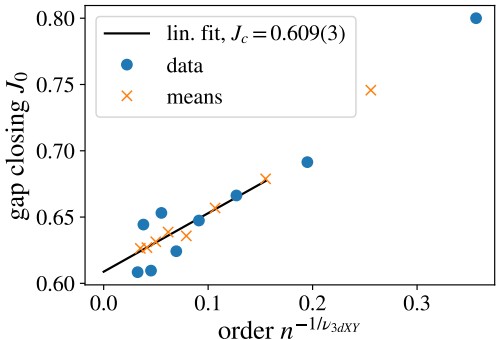

(b) Antiferromagnetic triangular TFIM, extrapolation of the critical point.

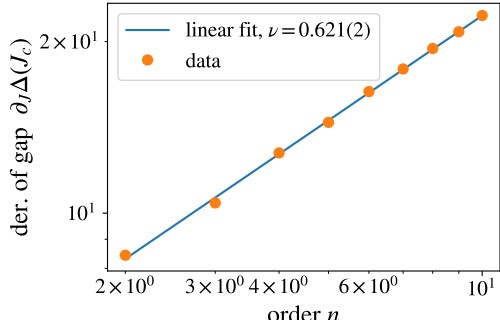

(c) Ferromagnetic triangular TFIM, extrapolation of the exponent.

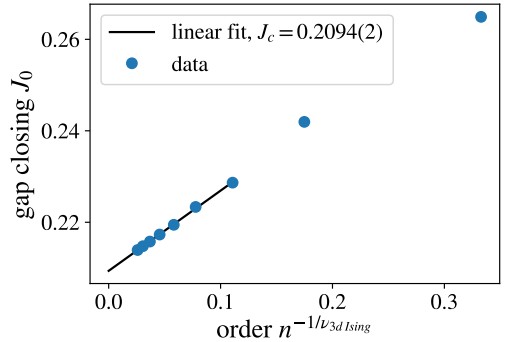

(d) Ferromagnetic triangular TFIM, extrapolation of the critical point.

Figure 2: Demonstration of the scaling relations for $\partial_J \Delta$ and $J_0$ for the antiferromagnetic (upper panel) and ferromagnetic (lower panel) TFIM on the triangular lattice. On the left-hand side fits for the exponents $\nu$ to the derivative of the energy gap, and on the right-hand side extrapolations for the critical point based on the closing of the gap are shown. For antiferromagnetic and ferromagnetic Ising interactions the results agree well with the expectation.

itself is not subject to the discretization $\delta$, but only to the discretization of the $\nu$ values for each of which a separate extrapolating fit can be performed to give a value for the critical point.

The discretization of the coupling $J$ is one limitation of the data. The spacing $\delta$ between the values of $J_i$ may limit the precision up to which the $\nu$ and the critical point $J_c$ can be extracted. We need the spacing for the frustrated TFIM $5 \times 10^{-3}$ and for the ferromagnetic TFIM $1 \times 10^{-3}$. The induced discretization on $\nu$ is typically of the order $10^{-3}$ and the re-induced discretization on the critical point is typically of the order $10^{-4}$. Another source of errors is the numerical derivative which is calculated as a finite difference with $h = \delta$ in all cases. The influence of the choice of $h$ has been checked to be small compared to the overall error of the results. We remark that it is conceptually not necessary to discretize the coupling. However, then a new deepCUT flow is required in every iteration which cannot be computed in parallel in contrast to a set of data points for predefined couplings.

Moreover, a source of uncertainty is the arbitrary choice of fit windows which has the largest influence on the results. For the critical point it has been found to be always beneficial to exclude the lowest orders from the fit as expected. For the exponent however, including all orders has been found to be more reliable for the models on the triangular lattice. In particular for the frustrated model, the choice of the fit window is crucial. Here, we can determine a reasonable choice by comparing to the literature values for $J_c$ and $\nu$. If the lowest order in the

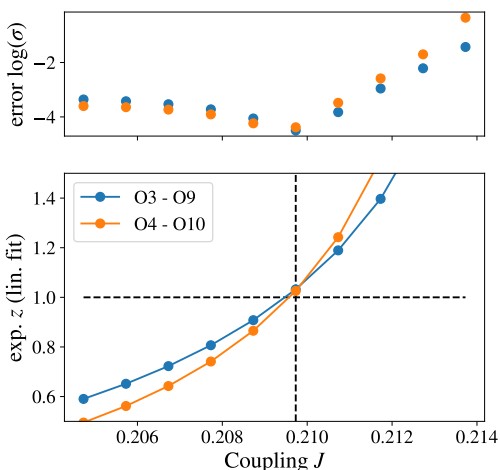

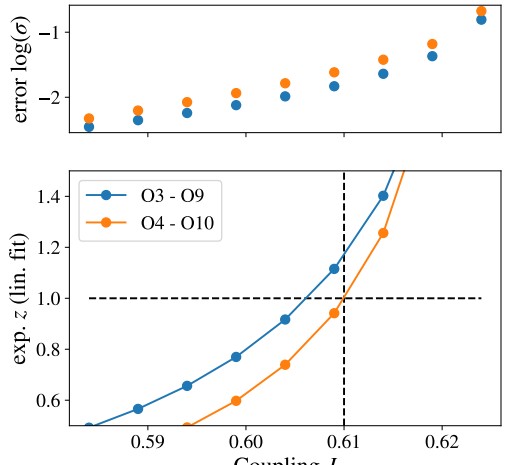

(a) Ferromagnetic TFIM on the triangular lattice, scaling of the energy gap with perturbative order.

(b) Antiferromagnetic TFIM on the triangular lattice, scaling of the energy gap with perturbative order.

Figure 3: Analysis of the correspondence of the length scale $\xi$ and the perturbative order $n$. For couplings $J$ in the vicinity of the critical point, a linear fit to the logarithm of the energy gap against the logarithm of the order is performed. At the critical point one finds that the fit errors (upper panels) for the well-behaved ferromagnetic model are minimal, implying that here the decay of the gap with order is well-described by a power law. For couplings away from the critical point the slope of the fit changes if the fit window is changed from order $3-9$ to $4-10$, in contrast. For both models at the respective critical point, indicated by a vertical dashed line, the exponent of the power law agrees well with $z = 1$ pointing towards $\xi \sim n$.

fit windows as in Fig. 2 are excluded additionally, one finds $J_c = 0.610(5)$ and $\nu = 0.71(5)$ for the frustrated TFIM, for example. This suggests errors in the order of $10^{-2}$ for the exponent and $10^{-3}$ for the critical point. These errors are more severe compared to those from the discretization and are expected to be the limiting errors of the scheme.

# 5 Sanity check for the correspondence of truncation and length scale

The gap is an important quantity for the analysis of quantum criticality since it displays much weaker corrections than the ground-state energy. The expectation for its scaling at the critical point

$$\Delta(J_c) \sim \xi^{-z}, \tag{32}$$

can be used to extract the critical exponent $z$ which is not useful to discriminate universality classes for the models we consider. However, here it can be used to check the scaling of the length scale $\xi$ with respect to the truncation parameter or perturbative order $n$. DeepCUT data of the energy gap around the critical point is used to fit a power law to the beavior of $\Delta$ against $n$. The resulting exponents for two fit windows [order 3(4) to order 9(10)] are plotted against the coupling $J$ for the ferromagnetic and antiferromagnetic TFIM on the triangular lattice in Fig. 3. For the ferromagnetic models one finds that the fit windows agree within good accuracy at the critical point on $z = 1$. Even from the fit errors it is apparent that the data is described best by a power law at the critical point. This evidence for $\Delta \sim n^{-1}$ is to be

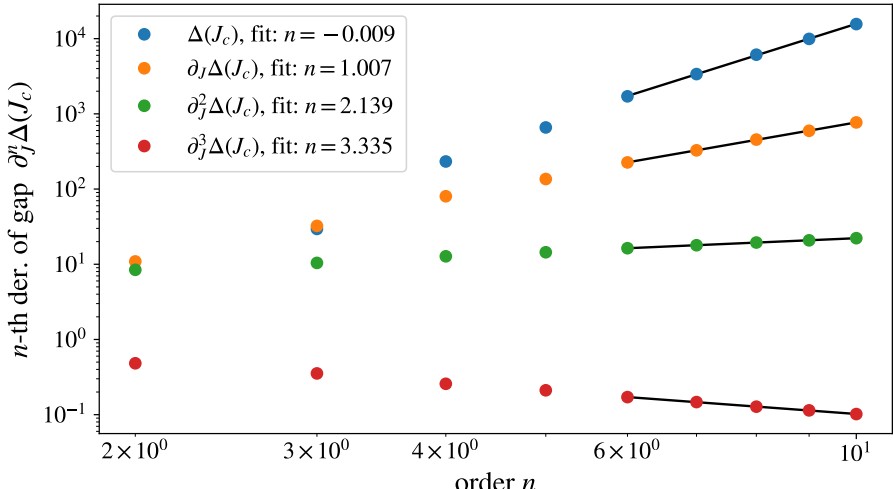

Figure 4: Scaling of the gap and the first three derivatives of the gap with length scale. Linear fits to the highest-order data points reveal values for the underlying exponent $-z + n/\nu$ from which we extract the values for $n$ which are given in the legend. The expectation of $n = 0, 1, 2, 3$ for the shown derivatives is met up to sufficient accuracy for the arguments made earlier which rely on the scaling of the $n$-th derivative of the energy gap.

compared to $\Delta \sim \xi^{-z}$ and the expectation $z = 1$ and confirms

$$n \sim \xi \,, \tag{33}$$

which is the foundation for the scheme presented in Section 4. For the antiferromagnetic TFIM on the triangular lattice the data is again disturbed by the oscillation of the gap with respect to even and odd orders which affects the fits for the exponent $z$. Still, $z = 1$ is found close to the literature value of the critical point.

We extend the analysis to derivatives of the energy gap. We have numerically stable access to the derivatives of the energy gap up to third order, and expect $\partial_J^n \Delta(J_c) \sim \xi^{-z+n/\nu}$. In Fig. 4 we plot the derivatives of the gap against the perturbative order for the ferromagnetic TFIM on the triangular lattice and extract an estimate for $n$ from a linear fit to the highest available orders. For the energy gap and its first derivative the error on the scaling behavior is negligible. For the second and third derivative the errors are in the range of 10%. At the same time we realize that generally, the magnitude of the derivatives decreases fast with the order of the derivative, reflecting the analyticity of the deepCUT data. Hence, corrections to the scaling relations are expected to only weakly impact a Taylor expansion of the deepCUT data about the critical point. That ensures that the assumption of the scaling form from Eq. 17 is valid in the context of deepCUT, since the scaling form can be derived solely from the scaling property of the energy gap and its derivatives.

# 6 Phase diagram of the frustrated TFIM bilayer

For the frustrated TFIM bilayer the energy gap is calculated for various configurations of the dimer parameters for the symmetric case $h := h_1 = h_2$

$$(J_\perp, h) \in \left\{ \left( \sin\left( \frac{j}{10} \frac{\pi}{2} \right), \cos\left( \frac{j}{10} \frac{\pi}{2} \right) \right), j = 0, \dots, 10 \right\} . \tag{34}$$

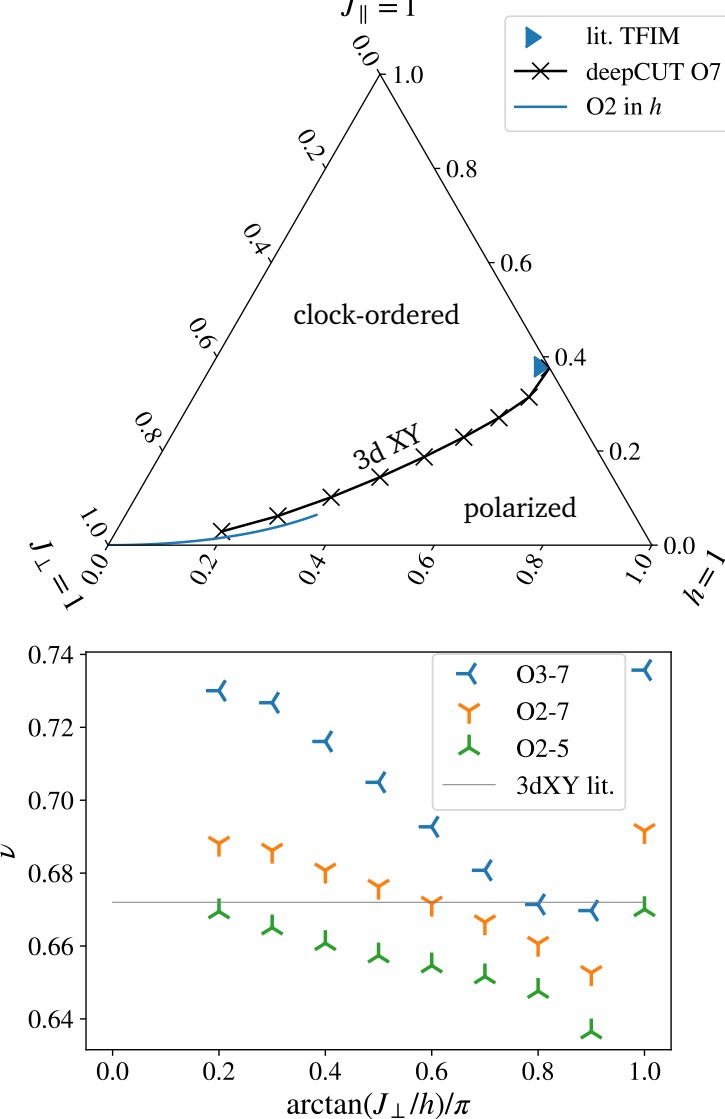

Figure 5: Quantum phase diagram and critical exponents of the frustrated TFIM bilayer. The phase diagram is a cut through the parameter space along the surface $J_{\parallel} + J_{\perp} + h = 1$. The right-hand side is the limit of decoupled layers, for which the numerically well known phase transition is indicated by the blue triangle, and the lower axis is the perturbative limit of decoupled dimers. Black crosses are the results for the critical point calculated with the iterative scheme. The blue line is the result for second order perturbation theory in the limit $J_{\parallel} = h = 0$. The exponents corresponding the each of the data points in the phase diagram are shown in the lower panel. Three fit windows are shown to capture the uncertainty of the result. The horizontal line is the literature value [28] for the transition.

The bare data is shown in Appendix E in Fig. 9. The results for $j = 0$ do not coincide with the frustrated single-layer TFIM. The differences are due to the degeneracy of the excitations in the layers 1 and 2 for $j = 0$ ($J_\perp = 0$), which cause the quasiparticle picture we use to break down. We still obtain results in that limit by considering one excitation infinitesimally shifted upwards energetically. However, this cannot be justified generally. For consistency with the earlier results however, we use the single-layer model as a substitute in that limit. The energy gap at all parameter settings is available up to order 7. As always for the energy gap, the 1:$n$-generator is used. In contrast to the single-layer models we find divergences in the data. They appear for $j \in \{0, 1, 8, 9, 10\}$. For $j = 0, 1$ order 6 diverges just before the closing of the gap which is likely due to an energetic overlap of qp blocks in the Hamiltonian. For $j = 8, 9, 10$ the divergences stem from the stiffness of the differential equation, which is induced by the separation of energy scales in the model in that limit.

The iterative scheme presented earlier is applied to all analyzable cases. The extrapolation of the critical point has been fitted to means of even and odd orders from order $3, 4$ to $6, 7$ and all orders are considered without averages for the exponent $\nu$. This is in analogy to the fit windows from the analysis of the frustrated single-layer TFIM in Fig. 2. For the case of decoupled layers the critical point is 4% lower than the literature value. Even though this result is not used but rather the result from the single-layer model which only deviates by 1%, it can be considered to suggest an order of magnitude of the error of the critical line. An estimate of the critical line for strongly coupled layers is given by the second-order expression from degenerate perturbation theory shown as solid blue line in Fig. 5. The observed discrepancy between our deepCUT result and the perturbative expression can be attributed to the low perturbative order.

For the critical exponent, the uncertainty is analyzed by considering multiple fit windows on the data at the critical point obtained from the iterative scheme applied with the full fit window from order 2 to 7. The data points at order 6 and 7 are the first ones which are strongly impacted by the alternation of orders. Therefore, only the fit windows $2 - 5$ and $3 - 7$ are reasonable to extract a trend for higher orders. The data suggests a tendency towards higher exponents. However, this may just be due to the onset of alternation. The data points are consistent with a critical line which is in the 3d-XY universality class throughout. In particular, no sign of a sudden change in the exponents is visible, which would point towards an intermediate phase on the other side of the critical line which does not exhibit an emergent $O(2)$ symmetry and is therefore not separated by a 3d-XY transition.

# 7 Conclusion

We have established a numerical scheme to detect and classify second-order quantum phase transitions for the real-space deepCUT method. As a first step, we have considered the ferromagnetic and antiferromagnetic single-layer TFIM on the triangular lattice to show that the scheme is able to distinguish between the 3d-XY universality class and the 3d Ising universality class. The success of the analysis relies on the strict agreement between the perturbative order, as the quantity controlling the truncation of deepCUT, and the length scale captured by the deepCUT data of that respective order. From the power law for the first derivative of the energy gap with respect to the control parameter, the critical exponent $\nu$ has been found to be accessible in a better way than it has been proposed in the literature for non-perturbative data for energy gaps [37]. A power law for the closing of the gap has been used to refine the previous extrapolation technique for deepCUT of fitting linear functions to the data versus the inverse order and has given access to values for the critical point in an accuracy competitive with the best data available for the frustrated TFIM on the triangular lattice [27,31]. To further

demonstrate the scheme, we have studied the quantum criticality of the TFIM bilayer on the triangular lattice. We have analytically derived two points at which a critical line of the 3d-XY universality sets in. In between, this critical line has been mapped out by applying the newly introduced iterative scheme. The obtained values for the critical exponents $\nu$ vary along the critical line but are consistent with the 3d-XY universality class within the expected uncertainty throughout. In particular, there is no sign of a sudden change of the nature of the transition which would have pointed towards an intermediate phase with a different second-order phase transition with other exponents. However, a first-order phase transition can not be excluded with our method.

The success of deepCUT in the quantum critical regime opens a wide range of possible further applications. Other observables than the energy gap can be calculated which also provides access to other critical exponents characterizing quantum phase transitions. Moreover, the low-field limit of the ferromagnetic Ising model is hard to extrapolate towards the quantum phase transition [6] which could be tackled with deepCUT, leaving room for an improvement in the results. Going to one-dimensional models such as the frustrated Ising ladder with single diagonal couplings in a transverse field, also other types of transitions like a Berezinskii–Kosterlitz–Thouless transition could be analyzed. This would require to adapt the iterative scheme since not the standard second-order power laws are expected anymore.

Importantly, one would also like to be capable of quantifying the errors on the critical point and the critical exponents better. This is intricate since the truncation errors and the freedom of the specifics of the fits involved in the iterative scheme all influence this error. In order to further increase substantially the precision of observables and critical exponents in the quantum critical regime, deepCUT is not expected to be the right method. Indeed, the truncation would have to be modified in order to capture longer length scales without requiring the calculation of all commutators necessary for quantum fluctuations in that range. This an interesting future direction to explore with CUT-based methods.

## Acknowledgments

We thank Max Hörmann, Patrick Adelhardt and Jan Koziol for useful discussions and Dag-Björn Hering for his contributions to the implementation. We thank Matthias Mühlhauser for providing NLCE data for the project. We thankfully acknowledge HPC resources provided by the Erlangen National High Performance Computing Center (NHR@FAU) of the Friedrich-Alexander-Universität Erlangen-Nürnberg (FAU).

**Funding information** We gratefully acknowledge financial support by the Deutsche Forschungsgemeinschaft (DFG, German Research Foundation) through projects SCHM 2511/13-1 (MRW/KPS) and through the TRR 306 QuCoLiMa ("Quantum Cooperativity of Light and Matter") - Project-ID 429529648 (KPS). KPS acknowledges further financial support by the German Science Foundation (DFG) through the Munich Quantum Valley, which is supported by the Bavarian state government with funds from the Hightech Agenda Bayern Plus.

# A Detailed Hamiltonians

In this appendix, we provide two more detailed and specific forms of the Hamiltonian of the frustrated TFIM bilayer (4). First, the full model can be rewritten as

$$
\begin{aligned}
H = \sum_{\mu} \sum_{a=1,2} \epsilon_a n_{a\mu} + J_{\parallel} \sum_{\langle \mu,\nu \rangle} \Bigg\{ & \left(\Gamma_+^2 + \Gamma_-^2\right)\left[b_{1\mu}^{\dagger}\left(b_{1\nu}^{\dagger} + b_{1\nu}\right) + \text{h.c.}\right] \\
& + \left(\Xi_+^2 + \Xi_-^2\right)\left[b_{2\mu}^{\dagger}\left(b_{2\nu}^{\dagger} + b_{2\nu}\right) + \text{h.c.}\right] \\
& - 2\Gamma_-^2\left[\left(n_{2\mu}b_{1\mu}^{\dagger}\left(b_{1\nu}^{\dagger} + b_{1\nu}\right) + (\mu \leftrightarrow \nu)\right) + \text{h.c.}\right] \\
& - 2\Xi_-^2\left[\left(n_{1\mu}b_{2\mu}^{\dagger}\left(b_{2\nu}^{\dagger} + b_{2\nu}\right) + (\mu \leftrightarrow \nu)\right) + \text{h.c.}\right] \\
& + 4\Xi_-^2\left[n_{2\mu}n_{2\nu}b_{1\mu}^{\dagger}\left(b_{1\nu}^{\dagger} + b_{1\nu}\right) + \text{h.c.}\right] \\
& + 4\Xi_-^2\left[n_{1\mu}n_{1\nu}b_{2\mu}^{\dagger}\left(b_{2\nu}^{\dagger} + b_{2\nu}\right) + \text{h.c.}\right] \\
& + \left(\Gamma_+\Xi_- + \Gamma_-\Xi_+\right)\left[\left(b_{1\mu}^{\dagger}\left(b_{2\nu}^{\dagger} + b_{2\nu}\right) + (\mu \leftrightarrow \nu)\right) + \text{h.c.}\right] \\
& - 2\Gamma_+\Xi_-\left[\left(n_{1\nu}b_{1\mu}^{\dagger}\left(b_{2\nu}^{\dagger} + b_{2\nu}\right) + (\mu \leftrightarrow \nu)\right) + \text{h.c.}\right] \\
& - 2\Gamma_-\Xi_+\left[\left(n_{2\mu}b_{1\mu}^{\dagger}\left(b_{2\nu}^{\dagger} + b_{2\nu}\right) + (\mu \leftrightarrow \nu)\right) + \text{h.c.}\right] \Bigg\}, \quad (\text{A.1})
\end{aligned}
$$

where $\mu \leftrightarrow \nu$ denotes a repetition of the previous term with indices $\mu$ and $\nu$ exchanged.

Second, for all numerical results, the symmetric Hamiltonian $H_{\text{sym}}$ with $h_1 = h_2$ is considered. It is obtained using the relations $u_- = 0$ and $v_- = 1$ which further imply $\Gamma_+ = \Gamma_- = v_+$ and $\Xi_+ = -\Xi_- = -u_+$. This significantly simplifies the Hamiltonian

$$
\begin{aligned}
H_{\text{sym}} = \sum_{\mu} \sum_{a=1,2} \epsilon_a n_{a\mu} + J_{\parallel} \sum_{\langle \mu,\nu \rangle} \Bigg\{ & 2v_+^2\left[b_{1\mu}^{\dagger}\left(b_{1\nu}^{\dagger} + b_{1\nu}\right) + \text{h.c.}\right] \\
& + 2u_+^2\left[b_{2\mu}^{\dagger}\left(b_{2\nu}^{\dagger} + b_{2\nu}\right) + \text{h.c.}\right] \\
& - 2v_+^2\left[\left(n_{2\mu}b_{1\mu}^{\dagger}\left(b_{1\nu}^{\dagger} + b_{1\nu}\right) + (\mu \leftrightarrow \nu)\right) + \text{h.c.}\right] \\
& - 2u_+^2\left[\left(n_{1\mu}b_{2\mu}^{\dagger}\left(b_{2\nu}^{\dagger} + b_{2\nu}\right) + (\mu \leftrightarrow \nu)\right) + \text{h.c.}\right] \\
& + 4u_+^2\left[n_{2\mu}n_{2\nu}b_{1\mu}^{\dagger}\left(b_{1\nu}^{\dagger} + b_{1\nu}\right) + \text{h.c.}\right] \\
& + 4u_+^2\left[n_{1\mu}n_{1\nu}b_{2\mu}^{\dagger}\left(b_{2\nu}^{\dagger} + b_{2\nu}\right) + \text{h.c.}\right] \\
& - 2v_+u_+\left[\left(n_{1\nu}b_{1\mu}^{\dagger}\left(b_{2\nu}^{\dagger} + b_{2\nu}\right) + (\mu \leftrightarrow \nu)\right) + \text{h.c.}\right] \\
& + 2v_+u_+\left[\left(n_{2\mu}b_{1\mu}^{\dagger}\left(b_{2\nu}^{\dagger} + b_{2\nu}\right) + (\mu \leftrightarrow \nu)\right) + \text{h.c.}\right] \Bigg\}. \quad (\text{A.2})
\end{aligned}
$$

# B Technical details of deepCUT

In this appendix, we review the technical details on the implementation of deepCUT based on the general derivation given in Sec. 3.

## B.1 Calculation of the operator basis and the mutual commutators

We have pointed out that epCUT flow equation (27) defines the truncation of deepCUT if it is reduced to its minimal form in the sense that only basis elements and contributions $D_{ijk}$ are kept which are strictly necessary to obtain the correct series expansion. Using the same basis and the same set of contributions in Eq. 25 yields exactly the desired deepCUT flow equation. As has been pointed out by Krull et al. [21], the implementation is intricate since the truncation can only be found after the full flow equation is set up, accordingly. If the full equation is infinite, a heuristically truncated epCUT flow equation is calculated instead, which is typically not minimal but guaranteed to be finite for a given perturbative order $n$. By maximally reducing that heuristic flow equation the deepCUT flow equation can be obtained as well. The algorithm to determine the basis of monoms and the set of contributions for the heuristic flow equation is presented hereafter. It iterates over orders $1 \ldots n$ and calculates commutators between basis elements which are directly stored in a tensor $D_{ijk}$. At the same time it extends the basis by newly arising monoms in every step.

Necessary inputs to the algorithm are the system Hamiltonian and the sought perturbative order $n$. The Hamiltonian $H(l = 0)$ is provided in the form

$$\{(A_i, O_{\min}(A_i)) | i = 0, \ldots, K\}, \tag{B.1}$$

where $K$ is the number of monoms occurring in the Hamiltonian at the beginning of the flow. The integer $O_{\min}(A_i) \in \mathbb{N}$ holds the information what is the leading order in the expansion

$$h_i(l, J) = \sum_{m=O_{\min}(A_i)}^{\infty} f_i^{(m)}(l) x^m, \tag{B.2}$$

for any monom $A_i$. It is stored during the computation since it later defines the maximally reduced flow equation. In the initial Hamiltonian it also defines which monoms are in $H_0$ (typically counting operators) having a minimal order of 0, and which are in $V$ (typically pair creation and annihilation and hopping terms) having an minimal order of 1. In order to store operators on lattice systems, translational invariance must be exploited. In general, symmetry-related monoms will at any point in the flow have equal prefactors. Hence, it is sufficient to only consider the symmetric superposition of all symmetry-related monoms as a representative of the corresponding symmetry class. Strictly, from all symmetries only the incorporation of translational invariance is a prerequisite for the algorithm since it reduces the starting Hamiltonian to finitely many monoms. E.g. for the TFIM the list of monoms and the corresponding minimal orders is initialized as follows:

| Index $i$ | Monom $A_i$ | $O_{\min}(A_i)$ |
|:---:|:---:|:---:|
| 0 | $\sum_i n_i$ | 0 |
| 1 | $\sum_{\langle ij \rangle} b_i^\dagger b_j^\dagger + \text{h.c.}$ | 1 |
| 2 | $\sum_{\langle ij \rangle} b_i^\dagger b_j + \text{h.c.}$ | 1 |

Starting from that list, the algorithm calculates mutual commutators of all monoms and adds new monoms to the list with their respective minimal order if they appear in the resulting commutators. We only consider models with particle-number-conserving $H_0$, implying

$$\hat{\eta}[H_0] = 0. \tag{B.3}$$

This is important since otherwise commutators from $[\hat{\eta}[H_0], H_0]$ may give rise to new monoms of minimal order 0. These monoms would again contribute to the commutator and the procedure would have to be iterated until no new monoms occur, which is in general not guaranteed

to take place. Then, custom truncation rules must be added for a self-consistent evaluation as for sCUT [18]. With the assumption in Eq. B.3, the lowest contributing monoms have minimal order 1 and stem from

$$[\hat{\eta}[V], H_0].\tag{B.4}$$

In general, also here multiple iterations to reach self-consistency may be necessary. However, if $H_0$ is local and the local Hilbert space is finite, which holds for all models in this article, self-consistency is guaranteed to be reached. After all, the first new monoms in this case stem from the commutator

$$[\hat{\eta}[V], V],\tag{B.5}$$

and have a minimal order of 2 and a larger spatial extent than the monoms of first order. For example for the TFIM on the square lattice the commutator

$$
\begin{aligned}
[\hat{\eta}[A_1], A_2] &= \left[ \sum_{\langle ij \rangle} b_i^\dagger b_j^\dagger - \text{h.c.}, \sum_{\langle ij \rangle} b_i^\dagger b_j + \text{h.c.} \right] \\
&= -2 \sum_{\langle\langle ijk \rangle\rangle} b_i^\dagger b_k^\dagger + \text{h.c.} + 4 \sum_{\langle\langle ijk \rangle\rangle} b_i^\dagger n_j b_k^\dagger + \text{h.c.},
\end{aligned}
\tag{B.6}
$$

yields several new monoms where double brackets $\langle\langle ijk \rangle\rangle$ denote next-nearest neighbors $i, k$ with intermediate site $j$. The prefactors on the right-hand side define the new contributions $D_{ijk}$. After all new monoms from the commutator in Eq. B.5 have been calculated, they are added to the list of monoms. The last contributions from order 2 stem from commutators of these new monoms with monoms of order 0. Again, no self-consistent evaluation is needed and no new monoms arise. Having identified all monoms with minimal order 2, the same procedure is repeated in order to find the monoms with minimal order 3. In general, monoms of order $m$ are calculated based on the monoms of all lower orders since the orders of $\eta$ and $H$ add up:

$$\partial_l H^{(m)}(l) = \sum_{p+q=m} [\eta^{(q)}(l), H^{(p)}(l)]\tag{B.7}$$

$$= [\eta^{(0)}(l), H^{(m)}(l)] + [\eta^{(1)}(l), H^{(m-1)}(l)] + \cdots + [\eta^{(m)}(l), H^{(0)}(l)].\tag{B.8}$$

All terms in the second line apart from the first and the last one can be calculated using only monoms of a minimal order strictly lower than $m$. The first term vanishes for all our models. The newly arising monoms only must be considered in the calculation of the last term $[\eta^{(m)}, H^{(0)}]$ which may require self-consistent iterations. If new monoms arise they are added to the list and the next order can be calculated. This structured way of going from low to high orders allows us to terminate the algorithm at any finite order since the number of monoms is guaranteed to be finite due to the bounded spatial extent of the monoms combined with the finite local Hilbert space [21]. The calculation can be speed up by shared-memory parallelization on the level of the individual commutators.

One caveat of the implementation is that each pair of monoms $A_i, A_j$ may contribute to two commutators. If a new monom $A_k$ arises in both commutators but with opposite sign, e.g. $D_{kij} = -D_{kji}$, it cancels out in the flow equation. For the correct truncation it is necessary to drop the monom and to use the properly increased minimal order for $A_k$. However, the sign of the contributions is ambiguous before the model parameters are fixed. We therefore omit that step in our implementation in order to calculate the commutators globally for all parameter values. Later, when the flow equation is set up, the sign is fixed and mutually cancelling contributions can be removed, but there is no correction of the potentially too low minimal order anymore. In that case our flow equation is no longer guaranteed to be minimal.

We expect the deviations as compared to the minimal flow equation small whatsoever, and the epCUT results are not affected at all. Summarizing, the necessary return values of the first step of the algorithm are the commutator symbols $D_{ijk}$ and the minimal orders for all monoms.

## B.2   Setting up the maximally reduced flow equation

The goal of the second step of the algorithm is to set up either the epCUT or deepCUT flow equation based on the information calculated in the first step. As pointed out in the derivation, the epCUT flow equation must be set up anyways in order to establish the correct truncation. We emphasize that the minimal flow equation is defined with respect to a targeted quantity. This is either the ground-state energy $E_0$ or the dispersion yielding the energy gap $\Delta$ within the scope of this article. The targeted quantity is given to the algorithm as a set of monoms which suffice to calculate the relevant matrix elements for the targeted quantity:

$$
\begin{aligned}
&\{\mathbb{1}\}, && \text{for } E_0, \\
&\{b_i^\dagger b_j + \text{h.c.} \,|\, |i-j| \le n\}, && \text{for } \Delta.
\end{aligned}
\tag{B.9}
$$

Additionally to the known minimal orders, one asks now for each monom what is the highest order $O_{\max}(A_i)$ which still influences any targeted quantity in order $n$ or lower. For the targeted quantities themselves this is straightforwardly $n$. For all other monoms the maximal order can be determined iteratively using the structure of the flow equation as encoded in $D_{ijk}$. This is done via the formal implicit definition

$$
O_{\max}(A_j) = \max_{\{i,k \,|\, D_{ijk} \ne 0 \vee D_{ikj} \ne 0\}} (O_{\max}(A_i) - O_{\min}(A_k)).
\tag{B.10}
$$

An initial set of maximal orders is defined as

$$
O_{\max}(A_i) =
\begin{cases}
n & \text{if } A_i \text{ targeted}, \\
0 & \text{else},
\end{cases}
\tag{B.11}
$$

which is plugged into the right-hand side of Eq. B.10. This increases the maximal orders monotonously in each iteration. The break-off condition is that the maximal orders do not change anymore which is guaranteed to take place for finitely many monoms at finite targeted order $n$. Then, the flow equation of epCUT

$$
\partial_l f_i^{(m)}(l) = \sum_{jk} \sum_{q+p=m} D_{ijk} f_j^{(q)}(l) f_k^{(p)}(l),
\tag{B.12}
$$

is set up with the restrictions

$$
\begin{aligned}
O_{\min}(A_i) &\le m \le O_{\max}(A_i), \\
O_{\min}(A_j) &\le q \le O_{\max}(A_j), \\
O_{\min}(A_k) &\le p \le O_{\max}(A_k).
\end{aligned}
\tag{B.13}
$$

This implies that monoms with $O_{\min} > O_{\max}$ are discarded, as well as contributions $D_{ijk}$ with $O_{\max}(A_i) < O_{\min}(A_j) + O_{\min}(A_k)$. The deepCUT flow equation is then uniquely reduced to all monoms and contributions which have not been discarded in the epCUT flow equation. This reduction step still can be done globally before the model parameters in $H_0$ are fixed. When the equation is set up with the remaining contributions the model parameters in $H_0$ must be fixed, since for each contribution $D_{ijk}$ it is checked whether the monom $j$ changes the energy according the the choice of parameters in $H_0$, otherwise the contribution is left out since we have $\eta[A_j] = 0$. Furthermore, if it changes the energy, $D_{ijk}$ is dressed with the appropriate

sign depending on whether the monom increases or lowers the energy. After this step is done, the flow equation is checked for cancellations of the form

$$D_{kij} = -D_{kji}, \tag{B.14}$$

which allow to reduce the equation further. In order to numerically solve the flow equation the perturbative parameter $x$ is fixed to determine the initial condition $H(l = 0) := H_0 + xV$.

## B.3 Efficient commutator algorithm using symmetries

The performance of the implementation heavily relies on using the symmetries of the physical model. Each monom shares its prefactor at any point in the flow with all monoms it can be transformed to by symmetry transformations, and it is more performant to track only a single prefactor for each set of symmetry-related monoms. Therefore, from now on, the term monom will refer to the symmetric superposition of itself with all local monoms in that set. These are created by omitting the sums for translational invariance and considering only relative positions to a reference site. Since this formal superposition is never lifted, the whole algorithm can be considered to act in the thermodynamic limit. The other symmetries are rotation and inversion symmetry, depending on the lattice, and the constraint of Hermiticity $A_i^\dagger = A_i$. These symmetries are implemented by storing the contributing local monoms explicitly. Algorithmically, all symmetries other than the translational invariance are implemented as functions which take a local monom and return a symmetric superposition of local monoms related by the symmetry. The global prefactor of the superposition is not necessarily one since a local monom can be mapped to itself by a symmetry transformation. These factors will be referred to as symmetry factors $s_i$ for each monom $A_i$. By the convention of Krull et al. the monoms are defined with prefactors of one such that formally the symmetry factors will not be considered part of the monom, implying

$$\text{sym}[a_i] = s_i A_i, \tag{B.15}$$

for any local representative $a_i$ of the symmetry class of the monom $A_i$.

At its heart, the deepCUT algorithm iteratively takes pairs of monoms $(A_j, A_k)$ and calculates their mutual commutator $[A_j, A_k]$. After normal-ordering the result can be expressed as

$$[A_j, A_k] = \sum_i D_{ijk} A_i, \tag{B.16}$$

which is known to be symmetric. Hence, it can be expanded in monoms by iteratively considering a contributing representative and identifying all symmetry-related terms in the result, removing them and storing the common prefactor in the tensor $D_{ijk}$. This procedure leaves plenty of room for optimizations. Introducing the corresponding symmetry factors, the commutator above can be rewritten as

$$[s_j A_j, s_k A_k] = \big[\text{sym}[a_j], \text{sym}[a_k]\big] = \text{sym}\Big[\big[a_j, \text{sym}[a_k]\big]\Big], \tag{B.17}$$

where $a_{j,k}$ again denote arbitrary local monoms of $A_{j,k}$. Due to the bilinearity of the commutator, one of the sums introduced by the sym-operations can be applied on the result of the commutation with a single representative. This saves exactly as many terms in the computationally costly process of normal ordering as generated by the symmetry operation. It is not even necessary to restore the result by applying the symmetries after the commutation. Instead, Eq. B.15 can be used to extract the structure coefficients from the unsymmetrized result. To see that, a monom $A_i$ is rewritten in terms of local monoms as $\sum_n a_i^n$, yielding

$$\big[a_j, \text{sym}[a_k]\big] = \sum_{i,n} c_i^n a_i^n. \tag{B.18}$$

It is noteworthy that many of the introduced coefficients $c_j^n$ are expected to be zero if $A_j$ consists of many representatives. Reinserting into Eq. B.17, one finds

$$[s_j A_j, s_k A_k] = \sum_{i,n} c_i^n \text{sym}[a_i^n] = \sum_{i,n} c_i^n s_i A_i. \tag{B.19}$$

Comparing this with Eq. B.16 directly implies

$$D_{ijk} = \frac{s_i}{s_j s_k} \sum_n c_i^n, \tag{B.20}$$

which allows for the computation of the structure coefficients involving only the symmetrization of a single monom if all symmetry factors are known.

Translational invariance is treated differently. The challenge is that the sums for translational invariance are not stored explicitly and therefore, any relative position between the two representatives must be considered [38]. For two representatives

$$A = A_{i+\delta_1} A_{i+\delta_2} \ldots A_{i+\delta_M}, \qquad B = B_{j+\delta_1'} B_{j+\delta_2'} \ldots B_{j+\delta_N'}, \tag{B.21}$$

with independent reference sites $i$ and $j$ and creation or annihilation operators $A_{i+\delta_n}$ on the sites $i + \delta_n$ for $n = 1, \ldots, M$ and for $B$ equivalently. The commutator $[A, B]$ can be rewritten using a product rule for commutators as

$$\sum_{m=1}^{M} \sum_{n=1}^{N} A_{i+\delta_1} \ldots A_{i+\delta_{m-1}} B_{j+\delta_1'} \ldots B_{j+\delta_{n-1}'} \left[ A_{i+\delta_m}, B_{j+\delta_n'} \right] B_{j+\delta_{n+1}'} \ldots B_{j+\delta_N'} A_{i+\delta_{m+1}} \ldots A_{i+\delta_M}. \tag{B.22}$$

The remaining commutator in each summand is a commutator of two creation or annihilation operators which is given by the commutation relation of the algebra directly, as for example for hardcore bosons

$$\left[ b_i, b_j \right] = \left[ b_i^\dagger, b_j^\dagger \right] = 0, \tag{B.23}$$

$$\left[ b_i, b_j^\dagger \right] = -\left[ b_i^\dagger, b_j \right] = \delta_{ij}(1 - 2n_i). \tag{B.24}$$

Importantly, the commutator in Eq. B.22 is either zero or proportional to a Kronecker symbol $\delta_{i+\delta_m, j+\delta_n'}$ which fixes the relative position of the reference site $i$ and $j$ by $j = i + \delta_m - \delta_n'$ for each summand. After the replacement the result is of the form

$$C = C_{i+\delta_1''} C_{i+\delta_2''} \ldots C_{i+\delta_K''}, \tag{B.25}$$

and still must be normal-ordered in order to be expanded in terms of normal-ordered monoms eventually.

## B.4 Simplification rules

In the calculation of commutators for epCUT and deepCUT an important optimization is the introduction of model-specific simplification rules [21]. They modify the algorithm to determine the operator basis and the mutual commutators presented in Section B.1 such, that it additionally takes the targeted quantity as an input and applies heuristics which to some extent avoid the computation of contributions and monoms which would later be discarded in the reduction of the differential equation system. Krull et al. distinguish between two kinds of simplification rules. *A posteriori* rules are applied after the calculation of a commutator

$$[\hat{\eta}[A_j], A_k] = \sum_k D_{ijk} A_i, \tag{B.26}$$

to all resulting monoms $A_i$ which are new at that point in the algorithm. Roughly speaking, if they contain too many creation or annihilation operators they cannot be reduced to a targeted quantity by repeated commutations with other monoms without exceeding the targeted order. In that sense, *a posteriori* rules give an upper bound for the maximal order of a monom depending on the targeted quantities and reduce the number of monoms stored during the computation and therefore also the total number of calculated commutators largely. The second kind of simplification rules are *a priori* rules. They are applied to a pair of monoms before their mutual commutator is calculated. They estimate the size of the smallest possible resulting monom and discard the computation of the commutator if that monom is too big. That does not reduce the number of monoms further, but saves the computation of a large fraction of commutators. In this section, simplification rules for all relevant models will be presented. For single-layer TFIMs they can be directly taken from earlier works on deepCUT [21,26]. For the bilayer TFIM modified versions of these rules have been found to be satisfactory. Henceforth, we take $n$ to be the targeted order. The number of creation or annihilation operators, which can at most be eliminated by a commutation with a monom of the perturbation $V$, is two for all our models. Furthermore, the number of targeted quasi-particles $q$ encodes whether the ground-state energy ($q = 0$) or the energy gap ($q = 1$) is targeted.

### B.4.1   A posteriori rules

The *a posteriori* rules formulated in Ref. [21] give

$$\widetilde{O}_{\max} = n - \left\lceil \frac{\max(c - q, 0)}{2} \right\rceil - \left\lceil \frac{\max(a - q, 0)}{2} \right\rceil \geq O_{\max}, \tag{B.27}$$

as an upper bound for the maximal order for a monom with $c$ creation operators and $a$ annihilation operators. For the single-layer TFIMs this bound has been observed to be sufficiently tight.

For the bilayer model in terms of two independent hardcore bosons per site the rule from Eq. B.27 holds, too. To make the bound tighter we make use of the the fact that a combination of a flavor-1 and a flavor-2 particle on neighboring sites like $b_{1\mu} b_{2\nu}$ cannot be cancelled by any term in the perturbation directly. This only happens in situations like the following

$$b_{1\mu} n_{2\mu} b_{2\nu} \cdot b_{1\mu}^\dagger b_{2\mu}^\dagger b_{2\nu}^\dagger = b_{2\mu}^\dagger + \dots, \tag{B.28}$$

where the first term stemming from the perturbation contains an additional counting operator and the second term is an example monom which is to be reduced to a targeted quantity. The prerequisite for this to happen is that the monom contains a site with a term $b_{1\mu} b_{2\mu}$ or its Hermitian conjugate. This is always fulfilled on a site with three operators or more or if the local operator is exactly $b_{1\mu} b_{2\mu}$ or the Hermitian conjugate. The rule is formulated as follows. If there is any site in the monom allowing for mixed cancellation according to the conditions described above, we apply the rule

$$\widetilde{O}_{\max} = n - \left\lceil \frac{\max(c_1 - q, 0) + c_2}{2} \right\rceil - \left\lceil \frac{\max(a_1 - q, 0) + a_2}{2} \right\rceil. \tag{B.29}$$

Else, we apply the following rule:

$$\widetilde{O}_{\max} = n - \left\lceil \frac{\max(c_1 - q, 0)}{2} \right\rceil - \left\lceil \frac{c_2}{2} \right\rceil - \left\lceil \frac{\max(a_1 - q, 0)}{2} \right\rceil - \left\lceil \frac{a_2}{2} \right\rceil. \tag{B.30}$$

Moreover, for all models an extended simplification rule can be applied as introduced for two-dimensional systems in Ref. [26]. The idea is that for example two operators $b_\mu b_\nu$ cannot be cancelled at the cost of one order if they are not nearest neighbors. For the extended

simplification rule the support of the monom is split into linked subclusters and by simple counting arguments a bound is determined, which is better than the ordinary bounds if the monom is sparsely distributed over the lattice. The rule has been taken over directly from Ref. [26] with a detailed explanation in the appendix.

### B.4.2  A priori rules

As mentioned before, *a priori* rules are applied to a pair of monoms $T, D$ of which the commutator is to be calculated. They give an upper bound for the maximal order of the resulting monoms. Krull et al. [21] find

$$\widetilde{O}_{\text{max},TD} = n - \left\lceil \max\left(\frac{c_T + c_D - \min(a_T, c_D)}{2} - q, 0\right)\right\rceil - \left\lceil \max\left(\frac{a_T + a_D - \min(a_T, c_D)}{2} - q, 0\right)\right\rceil, \tag{B.31}$$

with $c_D$ and $a_D$ being the number of creators and annihilators in the monom $D$. The sought upper bound is then given by

$$\widetilde{O}_{\text{max}} = \max\left(\widetilde{O}_{\text{max},TD}, \widetilde{O}_{\text{max},DT}\right). \tag{B.32}$$

Extended *a priori* rules reflecting the real space structure of monoms as also introduced in Ref. [21] have not been implemented but would be expected to yield further improvement. No adjusted rule for the bilayer model is used.

## C  Energy gap of TFIM on the square-lattice

This appendix presents results of the newly introduced scheme for the TFIM on the square lattice. This further demonstrates the generality of the scheme with respect to different geometries. The same analysis has been carried out for the TFIM on the triangular lattice in Sec. 4. Since the square lattice is bipartite we do not need to distinguish between ferromagnetic and antiferromagnetic coupling. In Fig. 6a the fit for the critical exponent $\nu$ is shown which exhibits similar accuracy as the ferromagnetic TFIM on the triangular lattice if the lowest two orders are neglected. The extrapolation of the critical point in Fig. 6b agrees well with the literature value 0.32850(10) [7] from series expansions. The sanity check for the energy gap is shown in Fig. 7.

Tab. 3 shows the result of the iterative scheme applied to the TFIM on the square lattice. As for the triangular lattice, the accuracy is sufficient to identify the expected 3d-Ising universality class. This proves that our scheme works well on different lattices.

## D  Analysis of the zero-temperature heat capacity

This appendix presents results on the critical exponent $\alpha$ which can be extracted from the ground-state energy by a similar argument as we have presented for the analysis of the exponent $z$ using the energy gap. The second derivative of the ground-state energy $\partial_J^2 E_0 \propto C/J^2$, which is proportional to the heat capacity $C$ around the critical point, is evaluated at the critical point using deepCUT. Here, its scaling

$$\partial_J^2 E_0 \sim n^{\frac{\alpha}{\nu}}, \tag{D.1}$$

is described by the critical exponent $\alpha$ assuming the truncation to be proportional to the length scale. However, other than for the energy gap, corrections to the scaling of the heat capacity

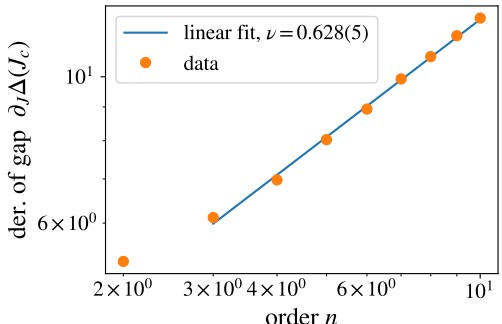

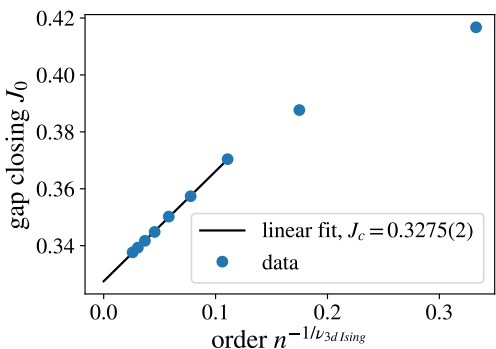

(a) TFIM on the square lattice, extrapolation of the exponent.

(b) TFIM on the square lattice, extrapolation of the critical point.

Figure 6: Demonstration of the scaling relations for $\partial_J \Delta$ and $J_0$ for the TFIM on the square lattice. On the left-hand side extrapolations for the critical point based on the closing of the gap, and on the right-hand side fits for the exponents $\nu$ to the derivative of the energy gap are shown.

must be expected, which can be described by

$$\partial_J^2 E_0 \sim A\xi^{\frac{\alpha}{\nu}} + B \,, \tag{D.2}$$

in the vicinity of the critical point as it has been done in Ref. [39]. The offset $B$ is expected to bias a log-log fit for the detection of a power law. For $\alpha > 0$, as for the 3d Ising universality class, one at least can expect asymptotic convergence since $(J - J_c)^{-\alpha}$ diverges at the critical point and the offset is suppressed relatively. However, for the 3d-XY universality class with negative $\alpha$, the value of the heat capacity at the critical point is finite and even the asymptotic limit is expected to be biased. For the case of the antiferromagnetic TFIM on the triangular lattice, the bias leads to an exponent with wrong sign and strongly overestimated absolute value, capturing the criticality not at all. Here, for both universality classes we therefore fit a function of the form

$$f(n) = a + bn^c \,, \tag{D.3}$$

with $c$ capturing the scaling behavior. The cost of doing that is that one parameter more must be fixed by the data as compared to a log-log fit.

Table 3: Iterations of the newly developed scheme to detect quantum-critical properties based on deepCUT data for the TFIM on the square lattice.

| Iteration | $\nu$ | $J_c$ |
|---:|---|---|
| 1 | 1 | 0.3147 |
| 2 | 0.688 | 0.3254 |
| 3 | 0.642 | 0.3271 |
| 4 | 0.635 | 0.3273 |
| 5 | 0.634 | 0.3273 |
| 6 | 0.634(4) | 0.3273(2) |
| literature | 0.630 | 0.3285(1) [7] |

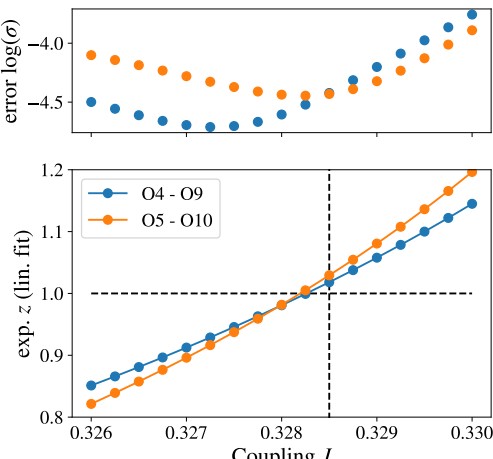

Figure 7: Scaling of the energy gap with the perturbative order for TFIM on the square lattice.

The results are shown in Fig. 8. In Fig. 8a the results for the frustrated TFIM on the triangular lattice are shown. Here, the series shows a strong even-odd effect. Importantly, the data is convergent and one can apply fitting techniques considering means between neighboring orders. The low and negative value of $\alpha = -0.01526(30)$ of the 3d-XY universality class is harder to capture than the 3d Ising value. The direct fit of the power law gives stable results for $\alpha$ of the order $-1 \times 10^{-3}$ with a trend towards values of larger magnitude. Even though this is no quantitative estimate of the exponent, it is a valuable result as it captures the feature of the exponent being small and negative. Generally, due to the offset, $\alpha$ is hardly detectable directly rather than using relations to other exponents. The ferromagnetic TFIM on the triangular lattice is shown in Fig. 8b. The resulting exponents seem to be scattered around the correct value, indicated by the horizontal gray line. Still, the scattering prohibits to extract a tendency to better convergence for higher orders. For the ferromagnetic TFIM on the square lattice the same analysis is carried out and presented in Fig. 8c. The difference is that one finds new contributions only at even orders as for the bare series. Hence, only few fit windows can be used reasonably. The quality of the exponents is comparable to those on the triangular lattice.

Overall, the ground-state energies of all models could be analyzed qualitatively. For the ferromagnetic models the fitting technique yields consistent results in contrast to a naive fit without offset. This is a strong indication that the offset is the dominant correction to the universal scaling behavior which makes the analysis of the heat capacity notoriously difficult.

# E    DeepCUT data for the bilayer TFIM

In this appendix we present the numerical deepCUT data which we have used to determine critical properties of the frustrated transverse-field Ising bilayer. In Fig. 9 the raw data is shown. The unit of energy is $\epsilon_+$ as defined in Eq. 12. In the limit of weak coupling for $j = 0, 1$ shown in Figs. 9a and 9b the energy gap of order 6 does not converge to the point where it presumably would close. Order 7 does not exhibit this problem which may be caused by the fact that it closes later due to the alternating behavior. In the limit of strongly coupled layers, the energy scales of the model are split into low-energy physics in the earlier introduced pseudospin degree of freedom and a large energy scale, which causes severe divergences in that limit as can be seen for $j = 9$, shown in Fig. 9f and $j = 10$ where no useful data is obtained



and no analysis is carried out. Surprisingly, for $j = 8$ the gap diverges in order 7 only in an intermediate regime and the closing of the gap is captured again, as can be seen in Fig. 9d. For all $j \in \{1, \ldots, 8\}$ vacancies in the data are filled by fitting a degree 15 polynomial to the 16 last data points which converged. The fits are represented by dotted lines. The result has been found to be robust against the changes in the order of the polynomial and the choice of which data points are used for the fit.

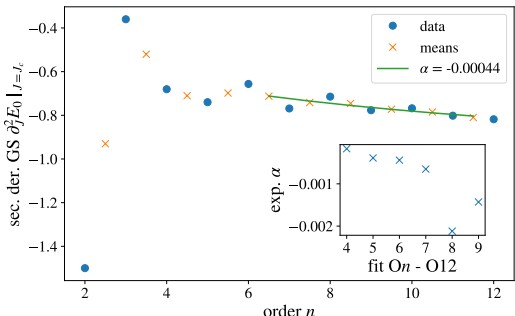

(a) Antferromagnetic TFIM on the triangular lattice.

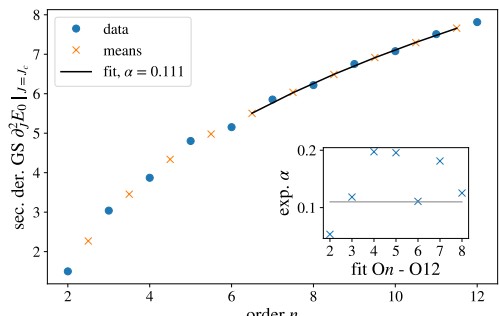

(b) Ferromagnetic TFIM on the trianglar lattice.

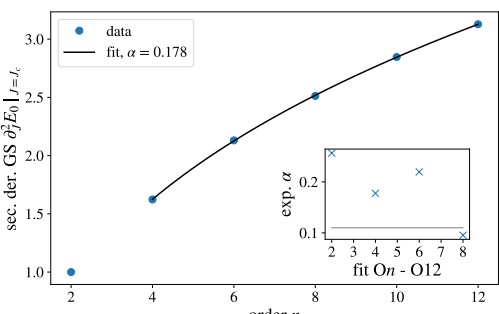

(c) Ferromagnetic TFIM on the square lattice.

Figure 8: Analysis of the ground-state energy at the critical point in TFIMs from deepCUT. The insets show the values for the exponent $\alpha$ from fit windows from order $n$ to 12. Literature values for $\alpha$ can be found in Tab. 1.

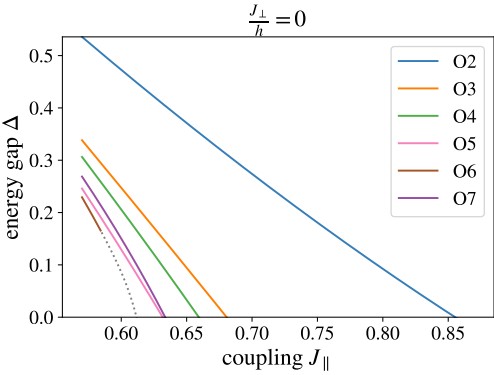

(a) Divergence in order 6 for $j = 0$. Single-layer data is used instead.

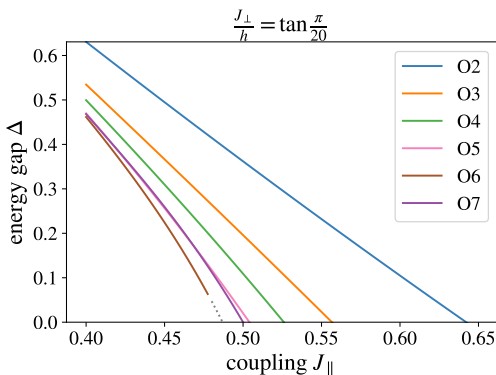

(b) Divergence in order 6 for $j = 1$. The data point may be influenced by the extrapolation.

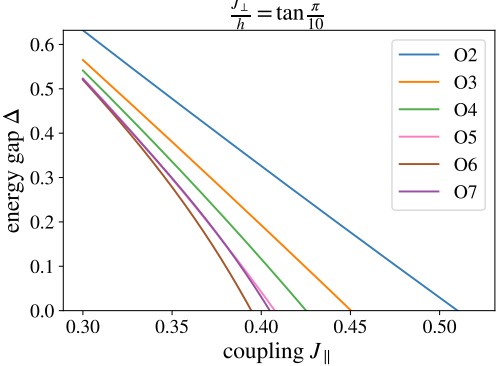

(c) For $j = 2$ we find the first set of parameters without divergences before the gaps close.

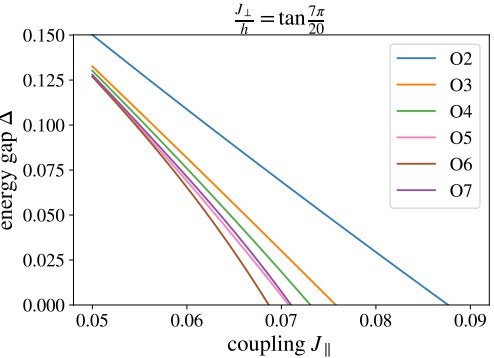

(d) For $j = 7$ we find the last convergent set of parameters.

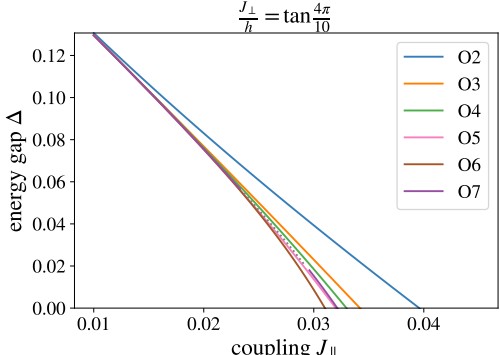

(e) An intermediate divergence in order 7 is found for $j = 8$.

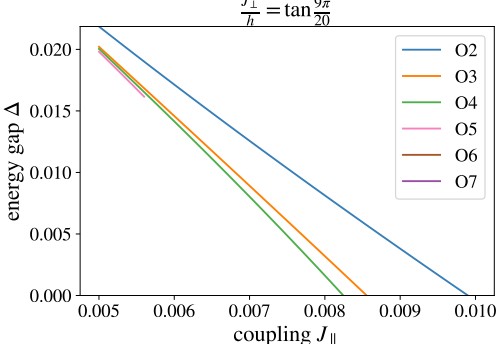

(f) Severe divergences starting from order 5 for $j = 9$. The data is not analyzed.

Figure 9: DeepCUT data of the energy gap for the frustrated transverse-field Ising bilayer. The intra-layer coupling is chosen as $J_\perp/h = \tan\left(\frac{j}{10}\frac{\pi}{2}\right)$ for $0 \le j \le 10$. Gray dashed lines indicate reconstructed data in regions where deepCUT diverges.

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
