# Peer review of "Extracting quantum-critical properties from directly evaluated enhanced perturbative continuous unitary transformations"

_SciPost Physics, doi:SciPost Phys. 17, 094 (2024)_

## Round 1 · Referee Report · Benedikt Fauseweh (Referee 1) · 2024-8-12

Report

The revisions made by the authors have addressed the previous concerns, and I am satisfied with the changes. I recommend the paper for publication. However, please note that there is a latex error on page 15 where a figure reference is not correctly set, which should be corrected before final publication.

Recommendation

Publish (easily meets expectations and criteria for this Journal; among top 50%)

---

## Round 1 · Referee Report · Anonymous (Referee 2) · 2024-8-13

Report

The authors have addressed my questions in the revised manuscript version. The paper is in good shape and I recommend it for publication.

Recommendation

Publish (easily meets expectations and criteria for this Journal; among top 50%)

---

## Round 1 · Referee Report · Anonymous (Referee 3) · 2024-8-27

Report

I am satisfied with the changes made to the manuscript in response to my report and those of the other referees. I recommend publication in Scipost.

One very small final point: The added sentence in the abstract should read "The data coincide with numerical evaluations of the truncated perturbative series and provide robust extrapolations beyond the perturbative regime".

Recommendation

Publish (meets expectations and criteria for this Journal)

---

## Round 1 · Author Response

Warnings issued while processing user-supplied markup:

  • Inconsistency: Markdown and reStructuredText syntaxes are mixed. Markdown will be used.
    Add "#coerce:reST" or "#coerce:plain" as the first line of your text to force reStructuredText or no markup.
    You may also contact the helpdesk if the formatting is incorrect and you are unable to edit your text.

Report of the First Referee -- Benedikt Fauseweh

We thank the author for the very positive evaluation of our manuscript and the recommendation for publication in SciPost Physics.

In the following we address the specific comments by the referee.

  • Abstract: "The data coincides with the perturbative series up to the order with respect to which the deepCUT is truncated." This sentence is a bit unclear. I propose "The data coincide with numerical evaluations of the truncated perturbative series and provides robust extrapolations beyond the perturbative regime."

Our response: We thank the referee for the suggestion. This has been changed in the revised version.

  • Page 3: It makes sense to also cite Phys. Rev. B 87, 184406 (2013) here, as it deals with the 1D TFIM with deepCUTs. The correspondence between truncation order to the correlation length was also discussed in my thesis: https://cmt.physik.tu-dortmund.de/storages/cmt-physik/r/uhrig/master/master_Benedikt_Fauseweh_2012.pdf

Our response: Thank you for making us aware of the reference. It is definitely relevant and we added it to our bibliography.

  • Page 10: The ROD is converged to 10^-9, why not double precision? If 10^-9 is reached it should already converge exponentially. Btw. There is a nice connection between speed of convergence and order of the calculation at the QCP, see also my thesis.

Our response: We sum up 10^6 contributions so that each have a numerical error of approximately 10^-15. Therefore 10^9 is in the ballpark of the worst case numerical error. In addition, we observed that all physical quantities are converged with sufficient accuracy. We have added both aspects on page 10 in the revised version to clarify this issue.

  • Page 11: I like the idea of the iterative scheme very much. So far, the scheme works with J values already calculated from the flow equation. Can this be generalized by including the CUT in the scheme by computing the solution of the flow equation for a given estimate for J_c and then obtain nu and then iterate? This avoids the need to discretize J with 10^-3 close to J_c. At most an additional point is required to compute the finite difference for the derivative.

Our response: This is an interesting suggestion that can certainly be implemented. For the models chosen in this paper, the computational cost of solving the flow equations is already substantial. Choosing a grid beforehand allows us to run all calculations in parallel, which we preferred in practice. We added a comment on page 12. Let us finally note that the discretization was not obstacle in our applications of the method which can be seen in the convincing values we obtained for the critical points and the corresponding exponents.

Page 12: It seems like the fit window is crucial for critical point and for the exponent AND they differ! This seems to me to be the most fine-tuned part of the method. Are there any ideas how to approach this problem systematically?

Our response: We thank the referee for this relevant question. We hesitate to include a full discussion of the fit window (and the associated error analysis) in our work, because with the calculated orders this is in our opinion beyond the scope of the current work, in particular for the alternating sequences present for the frustrated systems. We think it would be very interesting to design a system where higher orders can be determined so that this more refined analysis can be done in a proper fashion.

Fig 5: O2-7 and O2-5 seem to be more consistent but O3-7 not. Is this an even odd effect?

Our response: In general, it appears that taking higher orders into account leads to an enhanced value of the critical exponent when decreasing J_perp from right to left. In our opinion no even odd effect is clearly present. However, it is probably safer to say that the obtained orders are too low to make a definite statement about the behaviour of the critical exponent.

  • General: So far there are no error bars on the exponents. But from the fits, e.g. Fig 2 a) I would think the fitting algorithm gives you an estimate on the error. Is this smaller than 10^-3, so e.g. nu=0.687(..) ? Same question for J_c.

Our response: This is a valid point. The error of the fits are larger than 10^-3 for both, the unfrustrated and frustrated cases. In the revised version, we have therefore reduced the number of digits in the values of the critical points and the critical exponents by one. In addition, for the singled-layer cases we have added the error of the linear fits accordingly. Note that we have not done the same for the bilayer system because of the reduced order we have calculated.

Page 16: "In particular, there is no sign of a sudden change of the nature of the transition which could have pointed towards a first-order transition otherwise." Would a perturbative CUT with a 1:n generator be able to capture a first order phase transition anyway?

Our response: Thank you for the remark. We clarified that we cannot exclude a first order phase-transition and our statement only refers to the case with a second order phase-transition on page 17.

Page 16: "Importantly, one would also like to be capable of quantifying the errors on the critical point and the critical exponents better." Yes, I think that is the most important point. It is discussed a bit in the conclusion, but I would think that at least the effect of different fit windows could be discussed already quantitatively with the data available, without the need to carry out any new CUT calculations. Not necessarily in the main text but as an appendix.

Our response: This is a valid point. As stated above, we do not include a full discussion of the fit window in our work, because with the calculated orders this is in our opinion beyond the scope of the current work, in particular for the alternating sequences present for the frustrated systems. We think it would be very interesting to design a system where higher orders can be determined so that this more refined analysis can be done in a proper fashion.

Report of the Second Referee

We thank the author for the very positive evaluation of our manuscript and the recommendation for publication in SciPost Physics.

In the following we address the specific comments by the referee.

  • (1) The very technical discussion on the iterative procedure in the paragraph on page 11 starting with "One of the main sources of uncertainty..." was hard to follow for me. Since it seems that this iterative scheme is one of the main achievements in this work, I would like to motivate the authors to formulate it more clearly. The authors say that they start with nu=1 in their iterative scheme. How much do the results depend on this initial choice?

Our response: We have restructured and slightly rephrased this paragraph. We also stress that our results do not depend on the choice of nu. We have added a corresponding sentence in the revised version stating this explicitly.

  • (2) The caption of Fig. 5 mentions a dashed blue line. However I cannot see such a line in the figure.

Our response: We thank the referee for pointing out this type. We have erased the mentioning of the dashed blue line from the caption.

  • (3) On page 14 the authors say that their results in the decoupled bilayer limit do not agree with the single layer model. This sounds very strange since one would expect that the system is well behaved in the limit of small interlayer couplings. Can the authors comment on this in more detail or even eliminate this strange methodological artifact?

Our response: We agree that this is a counterintuitive finding which is caused by the fact that the quasiparticle picture we use for the bilayer model breaks down. The can be traced back to presence of degenerate excitation levels in the limit of two isolated layers and the quasi-particle operators we use to describe the corresponding excitations. As a consequence, we only obtain results by introducing an infinitesimal energy gap lifting the degeneracy. In the revised version, we have added two sentences to this matter.

(4) The blue line from perturbation theory deviates from the deepCUT results in Fig. 5 (top). More discussion about the reasons would be helpful. Is it because of the low order perturbation theory of the blue line or due to errors in deepCUT?

Our response: In our opinion this discrepancy originates from the low order of perturbation theory. We have added a corresponding discussion in the revised version of the article.

Report of the Third Referee

We thank the author for the very positive evaluation of our manuscript and the recommendation for publication in SciPost Physics.

In the following we address the specific comments by the referee.

    1. The review section is well referenced. But I found the description of the method in the paper itself not very pedagogical (both the main text and appendix B). One point that I found particularly opaque is central: how exactly the gap and its derivative ∂JΔ are computed. As a suggestion for the authors: I can imagine other readers would appreciate some more discussion especially of this last point.

Our response: Thank you for the remark. We add a paragraph on page 9 that explicitly states the final form of the Hamiltonian and how the dispersion and energy gap are computed. Note that the derivative of the gap is just calculated numerically.

    1. For the data in Fig. 2 I recommend including some measure of the goodness of fit to the computed points i.e. what are the errors on the exponent from this source? Errors are discussed at some points in the paper: for example coming from the choice of fitting window and in Fig. 3 and again in the lower panel of Fig. 5. Suggestion: could the authors bring together these contributions into a consolidated, quantitative discussion maybe at the end of the paper connecting to the remarks already in the discussion?

Our response: This is a valid point. Concerning the errors of the linear fits, we have now reduced the number of digits in the values of the critical points and the critical exponents by one. In addition, for the singled-layer cases we have added the error of the linear fits accordingly. We stress that this error for the frustrated case can be significantly reduced by taking averages over odd and even orders. Note that we have not done the same for the bilayer system because of the reduced order we have calculated. Concerning the fit window, we do not include a full discussion of the fit window in our work, because with the calculated orders this is in our opinion beyond the scope of the current work, in particular for the alternating sequences present for the frustrated systems. We think it would be very interesting to design a system where higher orders can be determined so that this more refined analysis can be done in a proper fashion.

    1. Fig. 5: the dashed blue line mentioned in the caption is not visible. If it overlaps with one of the other curves maybe change the plot style.

Our response: We thank the referee for pointing out this type. We have erased the mentioning of the dashed blue line from the caption.

    1. Fig. 8: I suggest including the known alpha exponents for 3D XY and Ising in the caption to this figure (or at least refer to table 1).

Our response.: We thank the referee for this comment. We have now included a link in the caption of figure 8 to the table 1 which cites the relevant references.

    1. General questions: Is it possible to capture a first order phase transition using this method? What about a higher order continuous transition like the KT transition?

Our response.: Our approach cannot capture a first-order phase transition. In the revised version on page 17, we have clarified this issue that we cannot exclude a first-order phase transition and that our statement only refers to the case of a second-order phase transition. Whether our approach is capable of detecting a KT-transition is an interesting but open problem. This is mentioned in the final conclusion and is left for future research.

---

## Round 1 · List of Changes

We have hihlighted all changes in red in the revised version. The details are given in our author comments specifically adressing the individual points and comments of the three referees.

---

## Editorial Decision

published